# SoftBinary Coding: A New Information-Theoretic Paradigm for Neural Compression via Fast Channel Simulation

Ezgi Ozyilkan [1] [*]   Sharang M. Sriramu [2] [*]   Elza Erkip [1]   Aaron B. Wagner [2]   Jona Ballé [1]

## Abstract

Neural compression is currently dominated by Nonlinear Transform Coding (NTC), which maps data to real-valued latents via continuous transforms. Despite its success, NTC suffers from train-test mismatch due to non-differentiable quantization, a "smoothness bias" inherent in continuous transforms that precludes optimality for certain sources, and a loss of "shaping gain" due to its use of scalar quantization. We propose SoftBinary Coding (SBC), an end-to-end learning paradigm that bypasses these limitations by using a stochastic binary latent space. In the spirit of vector quantization, SBC employs discrete representations and compresses them through a novel fast binary channel simulation scheme, for which we provide a proof of rate optimality. Experimental gains on information-theoretic sources address NTC's limitations both theoretically and practically, establishing discrete binary structures as a viable path toward reaching optimal rate–distortion bounds. Surprisingly, SBC also achieves state-of-the-art performance on vector quantization of i.i.d. sources, exceeding Trellis Coded Quantization of the Gaussian source.

## 1. Introduction

The current landscape of neural data compression is almost entirely defined by Nonlinear Transform Coding (NTC) (Ballé et al., 2017; Ballé et al., 2018; Minnen et al., 2018; Ballé et al., 2021). NTC operates by mapping the source data to a real-valued latent representation ($\mathbb{R}^L$) via learned neural transforms. To compress the data for transmission over the binary noiseless channel, NTC models rely on a "transform-quantize-entropy code" pipeline, where latent elements are quantized to integers ($\mathbb{Z}^L$) and then compressed via arithmetic coding. While originally conceived as an image compressor, the NTC framework has been successfully extended to a diverse array of modalities, including sources such as video (Rippel et al., 2019; Li et al., 2022), point clouds (Quach et al., 2019; Pang et al., 2022) and 3D Gaussian splats (Wang et al., 2024; Chen et al., 2024). While this approach has driven significant empirical success, at least three areas for improvement are identified in the existing literature:

i) The non-differentiability of hard quantization induces a mismatch between training and operational compression, necessitating surrogate differentiable approximations during training (Ballé et al., 2016; Theis et al., 2017; Agustsson & Theis, 2020). Alternatives such as the VQ-VAE formulation (van den Oord et al., 2017) are even worse in this regard—the training loss is entirely divorced from the operational rate, and entropy models need to be fit to the model post-hoc, with suboptimal results (Ozyilkan et al., 2023).

ii) The use of continuous transforms introduces an inherent "smoothness bias" (Bhadane et al., 2022; Ozyilkan et al., 2024a), limiting the model's ability to represent the sharp, discontinuous structures required to achieve optimal rate–distortion performance for certain sources. This is revealed by comparing compressors trained on information-theoretic sources (Wagner & Ballé, 2021; Bhadane et al., 2022) against theoretical optima.

iii) Realizing *shaping gains* (Zamir, 2014) via high-dimensional vector quantization (VQ) is computationally prohibitive (Gray, 1984); NTC schemes are typically restricted to scalar quantization (Ballé et al., 2021), thereby leaving performance gains on the table relative to the optimal asymptotic rate–distortion tradeoff (Lei et al., 2025).

The latter two points are particularly relevant in the application of NTC to *non-image* sources: Image compression tends to focus on the high-rate regime, where densities may be assumed smooth and shaping gains are negligible. Optimally compressing data such as sparse point cloud attributes may benefit from more flexible schemes.

[*]Equal contribution [1]Tandon School of Engineering, New York University, Brooklyn, NY [2]Cornell University, Ithaca, NY. Correspondence to: Ezgi Ozyilkan <ezgi.ozyilkan@nyu.edu>, Sharang M. Sriramu <sms579@cornell.edu>.

*Proceedings of the 43rd International Conference on Machine Learning*, Seoul, South Korea. PMLR 306, 2026. Copyright 2026 by the author(s).

NTC (dithered quantization):
$$\mathcal{F} = \mathcal{U}\big(Z \mid V - \tfrac{1}{2}, V + \tfrac{1}{2}\big)$$

SoftBinary Coding:
$$\mathcal{F} = \text{Bern}\big(Z \mid \tfrac{1+V}{2}\big)$$

*Figure 1.* Training scheme for learning-based lossy neural compression with channel simulation. The neural network $f_\theta$ produces parameters $V$ of the encoder distribution, a parametric family $\mathcal{F}(Z \mid V)$. The latent representation $Z$ is a sample from this family (operationally, channel simulation produces the sample at the decoder). The encoder $q_\theta$ and prior $p_\psi$ (a model of the marginal distribution of $Z$) evaluated over $Z$ determine the training rate $R = \mathbb{E}[D_{\text{KL}}(q_\theta \| p_\psi)]$, while the decoder network $g_\phi$ reconstructs the input as $\hat{X}$ with training distortion $D = \mathbb{E}[d(X, \hat{X})]$, optionally utilizing side information $Y$ (in the distributed compression setup).

In this work, we propose *SoftBinary Coding* (SBC), an alternative framework for end-to-end neural compression that directly addresses these limitations. i) SBC replaces deterministic quantization and entropy coding with *channel simulation* (Li, 2024), resulting in a training objective that matches the rate–distortion criterion evaluated at test time. ii) At the same time, it restricts the latent space to binary variables $\{0,1\}^L$ which eliminates the smoothness bias induced by continuous-valued transforms, and iii) employs a scalable high-dimensional scheme that achieves the shaping gains associated with VQ.

Our scheme depends on a confluence of developments in different fields. First, we rely on `PolarSim`, a scalable binary-output channel simulator by Sriramu et al. (2024). It applies the polar transform to several independent binary output channels, inducing polarized *subchannels* (see the discussion in Sec. 3.1 to follow) which admit efficient simulation schemes. `PolarSim`, however, requires that the binary outputs have uniform marginals, but we find that *nonuniform* marginals are needed here, as the trained model would otherwise not have enough flexibility to obtain the desired performance. We thus extend `PolarSim` to allow for nonuniform outputs and indeed independent but not identically distributed channels. We prove that this extension is rate-optimal. Second, for end-to-end training, we rely on the *VarGrad* method of Richter et al. (2020), which provides low-variance unbiased gradient estimates for latent-variable models with discrete latents such as ours. Our results demonstrate that this integrated scheme not only performs well empirically but also establishes SBC as a viable path forward for neural data compression.

## 2. Background

### 2.1. Lossy neural data compression

Nonlinear Transform Codes learn compressible representations $Z$ of the source $X$; they resemble probabilistic latent-variable models such as Variational Autoencoders (VAEs; Kingma & Welling, 2014). The goal is to minimize a joint rate–distortion objective over model parameters $\theta, \phi, \psi$:

$$L(\theta, \phi, \psi) = R + \lambda D. \tag{1}$$

The distortion $D$ measures the fidelity between $X$ and $\hat{X}$ (the source as reconstructed from $Z$), typically via mean squared error, though perceptual or realism-based metrics are becoming increasingly popular (Mentzer et al., 2020; Ballé et al., 2025). The rate $R$ gives the expected code length in bits necessary to communicate the representation $Z$.

More concretely, neural compressors consist of three parameterized components: an encoder distribution $q_\theta$, a decoder $g_\phi$, and an entropy model or prior $p_\psi$ (Figure 1). The encoder maps $X$ to a latent representation $Z$, which is then reconstructed by the decoder as $\hat{X} = g_\phi(Z)$.

The two components in (1) are generally of the form:

$$R = \mathbb{E}_{\substack{X \sim p_X \\ Z \sim q_\theta(Z|X)}} [\log q_\theta(Z \mid X) - \log p_\psi(Z)] \tag{2}$$

$$= \mathbb{E}_{X \sim p_X}[D_{\text{KL}}(q_\theta(Z \mid X) \parallel p_\psi(Z))],$$

$$D = \mathbb{E}_{\substack{X \sim p_X \\ Z \sim q_\theta(Z|X)}} [d(X, g_\phi(Z))]. \tag{3}$$

Taken together, the terms resemble the training loss of a VAE, where the likelihood term of the VAE is replaced by the distortion term weighted with a hyperparameter $\lambda$, determining the trade-off between compression and reconstruction fidelity (Ballé et al., 2017).

In NTC, $q_\theta$ is a standard uniform distribution with center $V$; sampling from $q$ thus corresponds to adding independent uniform noise to $V$ (Ballé et al., 2017). This induces a rate–distortion objective that is both amenable to gradient-based optimization and admits exact operational realization. It is differentiable, such that an unbiased estimate of the gradient of the loss with respect to the encoder parameters $\theta$ can be obtained using Monte Carlo sampling (a trivial special case of the "reparameterization trick"). It can be achieved operationally using arithmetic coding and *dithered quantization* (Zamir & Feder, 1992). Here, the quantization offset (the dither) is randomized uniformly using a pseudo-random number generator with shared seed (Roberts, 1962;

Schuchman, 1964).

Assuming the arithmetic coder operates on a sufficiently large block, grouping many samples together to maximize efficiency, $R$ is essentially the expected code length of the compressed representation (Cover & Thomas, 2006): The first term in (2) is constant and equal to 0, and the second amounts to the cross-entropy between the marginal distribution of latents and the prior (entropy model) $p_\psi$.

In practice, the dither is usually fixed after training, resulting in an entropy-coded scalar quantization (ECSQ) method, which can achieve better results than dithered quantization. However, this deviates from the framework presented here, as it makes the encoder deterministic. In that case, the operational rate and distortion are:

$$R_{\text{ECSQ}} = \mathbb{E}_{X \sim p_X}[-\log_2 p_\psi(\lfloor f_\theta(X) \rceil)], \tag{4}$$

$$D_{\text{ECSQ}} = \mathbb{E}_{X \sim p_X}[d(X, g_\phi(\lfloor f_\theta(X) \rceil))], \tag{5}$$

where $\lfloor \cdot \rceil$ denotes rounding to the integer grid $\mathbb{Z}^n$ shifted by the fixed dither. Due to this operation, the operational loss function is unfortunately not differentiable. There is thus necessarily a disconnect between the training loss and the operational loss. Finding the optimal fixed dither post-training may require ad-hoc methods. Agustsson & Theis (2020) propose an annealing procedure to interpolate between (1) and the operational loss during training.

This framework can be generalized by selecting encoder distributions $q_\theta$ from larger parametric families $\mathcal{F}$, and there are potential advantages to doing so. While gradient estimators may exist for the resulting model, making it possible to train it, operationally implementing it requires a rate-efficient method to *simulate* the *channel* $q_\theta(Z|X)$; i.e., to enable the decoder to draw a sample $Z$ without knowing $X$.

### 2.2. Channel simulation

*Channel simulation* is a paradigm for communicating samples drawn from a noisy channel using a rate-efficient description. It can be interpreted as a "soft" generalization of entropy-coded quantization, where randomness replaces deterministic quantization decisions. We provide a formal definition below (using the information-theoretic convention of indicating the length of vectors by superscripts).

Consider the sequence of independent random variables

$$(V_i, Z_i) \sim P_{V_i, Z_i}, \text{ for } i \in \{1, \ldots\}, \tag{6}$$

where each $P_{V_i, Z_i}$ is a probability measure on $\mathcal{V} \times \mathcal{Z}$, and $P_{V_i}$ denotes the corresponding marginal on $\mathcal{V}$. We refer to the conditionals $P_{Z_i|V_i}$ as the *target channels*. Define the *common randomness* to be a random variable $S \sim P_S$ that takes values in some arbitrary $\mathcal{S}$, is independent of the *source sequence* $V^k$ for all $k \in \mathbb{N}$, and is infinitely divisible.

For a given block length $N = 2^n$, for some $n \in \mathbb{N}$, the *encoder* observes the source realization $V^N$ along with the common randomness $S$ and outputs a prefix-free bit string $\mathsf{Enc}(V^N, S)$, which it transmits losslessly to the decoder. The *decoder* uses this message in conjunction with the common randomness to produce a reconstruction $Z^N = \mathsf{Dec}(\mathsf{Enc}(V^N, S), S)$ such that $(V^N, Z^N) \sim \prod_{i=1}^{N} P_{V_i, Z_i}$.

Our goal is to design a scheme, consisting of the common randomness, the encoder, and the decoder, which minimizes the average amortized length of the encoder's message $\frac{\ell(\mathsf{Enc}(\cdot))}{N}$. The smallest asymptotically achievable *rate*, i.e., the average amortized communication cost, is known (Li & El Gamal, 2018) to be the *mutual information* $\frac{1}{N} \sum_{i=1}^{N} I(V_i; Z_i) = \frac{1}{N} \sum_{i=1}^{N} D_{\text{KL}}(P_{Z_i|V_i} \| P_{Z_i})$. In the context of neural compressors, this cost is a function of the network parameters (see (2)) and is directly used as part of the training objective.

Dithered quantization (Sec. 2.1) can be viewed as a scheme for simulating a particular channel, namely one with uniform additive noise (Zamir & Feder, 1992). We improve upon NTC by replacing dithered quantization with a more powerful channel simulator. Existing general-purpose channel simulation schemes (Li & El Gamal, 2018; Flamich, 2023; Phan & Khisti, 2025) are based on random coding and have exponential computational complexity in the block length $N$, rendering them impractical. However, by restricting our model to use binary latents, i.e., $\mathcal{Z} = \{0, 1\}$, we are able to utilize scalable schemes from the error-correcting code literature. In particular, *polar codes* (Arıkan, 2009) have shown promising performance in different regimes of channel simulation (Chou et al., 2018; Sriramu et al., 2024).

## 3. Simulation of independent binary-output channels

SBC uses `PolarSim` (Sriramu et al., 2024), an asymptotically rate-optimal scheme for simulating binary-output channels with pseudolinear computational complexity. In its original form, `PolarSim` is limited to i.i.d. channels with binary, uniform outputs. Since the latents that emerge from the trained model are not guaranteed to be identically distributed, as noted in the introduction, we extend `PolarSim` to allow for independent, but not necessarily identically-distributed, channels with arbitrary marginal distributions for the output bits. We develop a generalization of Arıkan's source polarization theorem (Arıkan, 2010) and use it to prove first-order optimality of the generalized `PolarSim` scheme. While source polarization has been generalized to different source models (Guruswami et al., 2018; Şaşoğlu & Tal, 2019), the non-stationary memoryless regime is underexplored (but see Mahdavifar (2020) for a channel cod-

ing variant). In this section, as in Sec. 2.2, we follow the information-theoretic convention of indicating the length of vectors by superscripts.

### 3.1. Simulating polarized channels

Sriramu et al. (2024, Sec. 3.1) describes the following simulation scheme for simulating a binary output channel $P_{Z|V}$:

1. Generate $S \sim \mathrm{Unif}[0, 1]$ using common randomness.

2. Upon observing a realization $v \sim V$ at the encoder, compute $Z = \mathbf{1}\{S > P_{Z|V}(0|v)\}$.

3. Compute $\tilde{Z} = \mathbf{1}\{S > P_Z(0)\}$ at both the encoder and the decoder.

4. Transmit the difference $\Delta = Z \oplus \tilde{Z}$ to the decoder after lossless compression, where $\oplus$ denotes the XOR operation, so that $Z = \Delta \oplus \tilde{Z}$ can be recovered.

Sriramu et al. (2024) assume that $P_Z$ is uniform, but the scheme is easily extended to the general case above. Even for channels with uniformly distributed outputs, this simple scheme is suboptimal in general in the sense that its rate, which is approximately $H(\Delta)$ if the overhead due to the lossless compressor is amortized over several runs, exceeds the mutual information lower bound $I(Z; V)$ (Sriramu et al., 2024, Appendix A). However, if the channel is *polarized*, i.e., if the mutual information $I(Z; V)$ is close to either 0 or 1, Sriramu et al. (2024, Appendix A) shows that the gap between $H(\Delta)$ and $I(Z; V)$ vanishes.

If we remove the uniformity assumption on $P_Z$ and consider a larger class of source–channel pairs, one additional polarized regime becomes possible: channels with near-deterministic output. In this regime, the same correction-based scheme is expected to be near-optimal, since near-determinism leaves little room for $P_{Z|V}$ to differ substantially from $P_Z$ with high $P_V$-probability.

This suggests that polarizing transforms, which replace the target channel with polarized *subchannels*, in conjunction with Sriramu et al. (2024)'s simulation scheme, can lead to efficient channel simulation algorithms even for channels with non-uniform output marginals. In the following subsections, we define such a polarizing transform and the resulting simulation algorithm, and prove that it is asymptotically rate-optimal.

### 3.2. Source polarization

Our construction is based on a modified version of the source-polarizing transform proposed by Arıkan (2010). We first review the basic $2 \times 2$ transform that underlies the construction.

Let $Z_1 \sim \mathrm{Bern}(p_1)$ and $Z_2 \sim \mathrm{Bern}(p_2)$ be independent, with $p_2 < p_1 < \frac{1}{2}$. Define the invertible transform

$$(U_1, U_2) \triangleq (Z_1 \oplus Z_2, Z_2). \tag{7}$$

Since this mapping is bijective, the joint entropy is preserved:

$$H(Z_1) + H(Z_2) = H(U_1, U_2) = H(U_1) + H(U_2 \mid U_1).$$

However, the entropies of the individual components become more extremal: $H(U_1) > H(Z_1)$, since the XOR operation mixes $Z_1$ with the additional uncertainty in $Z_2$, thereby increasing the marginal uncertainty of $U_1$ (and correspondingly decreasing $H(U_2 \mid U_1)$).

Arıkan (2010) exploits this mechanism recursively for $N$ i.i.d. sources by repeatedly applying the same entropy-splitting operation $n$ times. He shows that the induced conditional entropies *polarize*, so that the fraction of transformed variables whose conditional entropies remain bounded away from both 0 and 1 vanishes asymptotically.

Theorem B.1 shows an analogous result in the non-stationary setting. To facilitate analysis, it uses a modified version of Arıkan's polar transform (Arıkan, 2010), in which a random permutation is applied before each recursive stage, and the number of recursion levels is fixed at $n^*$, which need not equal $n$. The modified transform, called `PermutedPolarTransform`, is described in Appendix A. The proof of Theorem B.1 appears in Appendix B.1.

### 3.3. Algorithm for channel simulation

We will apply `PermutedPolarTransform` to the channel outputs $Z^N$, with the random seed $\Pi$ obtained from an independent split of the common randomness $S$, and $n^* \in \{1, \cdots, n\}$ chosen according to Theorem B.1 obtaining

$$U^N = \texttt{PermutedPolarTransform}(Z^N, \Pi, n^*). \tag{8}$$

This transforms the problem of simulating $N$ independent target channels $V_1 \to Z_1$, $V_2 \to Z_2$, ..., $V_N \to Z_N$ into the problem of simulating the dependent *subchannels* $(V^N, \Pi) \to U_1$, $(V^N, U_1, \Pi) \to U_2$, ..., $(V^N, U^{N-1}, \Pi) \to U_N$.

Theorem B.1 guarantees that, for all but a vanishing fraction of indices $i$, both $H(U_i \mid U^{i-1}, V^N, \Pi)$ and $H(U_i \mid U^{i-1}, \Pi)$ are close to either 0 or 1 (in the latter case, one takes $V^N$ to be null). Consequently, the corresponding mutual informations,

$$\begin{aligned} I(U_i; V^N \mid U^{i-1}, \Pi) \\ = H(U_i | U_1^{i-1}, \Pi) - H(U_i | U_1^{i-1}, V^N, \Pi), \end{aligned} \tag{9}$$

are also polarized. We will then use Sriramu et al. (2024)'s

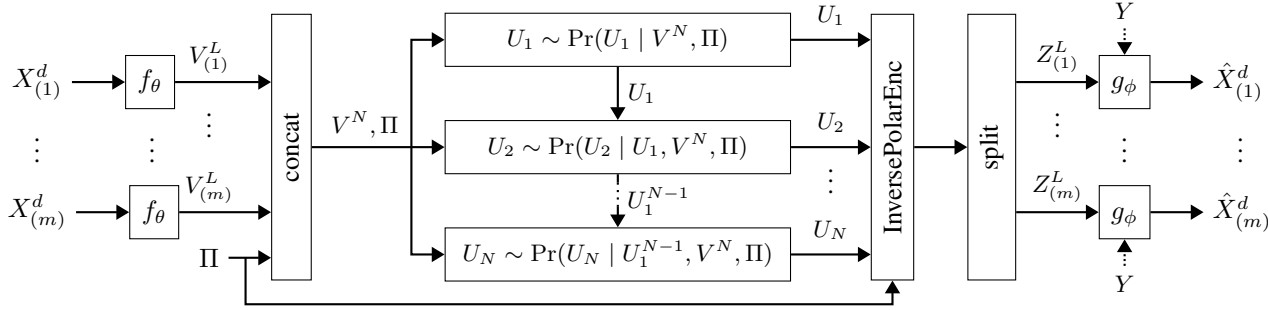

*Figure 2.* System diagram for the operational scheme using channel simulation (Sec. 3). We concatenate the encoded messages $V_{(i)}^L = f_\theta(X_i)$ for several independent source realizations and use Algorithms 1 and 2 to generate the latent samples $Z^N$ at the decoder.

---

**Algorithm 1** Generalized PolarSim: Encoder side

---

**Input:** Block length $N$
**Input:** Source sequence $v^N \sim \prod_{i=1}^N P_{V_i}$
**Input:** Random seed $s^N \overset{\text{i.i.d.}}{\sim} \text{Unif}(0,1)$
**Input:** Permutation randomness $\Pi \in \mathcal{S}$
**Output:** Compressed bit string $b \in \{0,1\}^*$
 1: **for** $i = 1, \ldots, N$ **do**
 2:      $P_i \leftarrow \Pr(U_i = 0 \mid U^{i-1} = u^{i-1}, \Pi)$
 3:      $Q_i \leftarrow \Pr(U_i = 0 \mid U^{i-1} = u^{i-1}, V^N = v^N, \Pi)$
 4:      $\tilde{u}_i = \mathbf{1}\{s_i > P_i\}$
 5:      $u_i = \mathbf{1}\{s_i > Q_i\}$
 6:      $\Delta_i \leftarrow u_i \oplus \tilde{u}_i$
 7: **end for**
 8: $b \leftarrow \text{COMPRESS}(\Delta^N)$
 9: **return** $b$

---

**Algorithm 2** Generalized PolarSim: Decoder side

---

**Input:** Block length $N$
**Input:** Compressed bit string $b \in \{0,1\}^*$
**Input:** Shared randomness $s^N \overset{\text{i.i.d.}}{\sim} \text{Unif}(0,1)$
**Input:** Permutation randomness $\Pi \in \mathcal{S}$
**Input:** Probability table $\overline{P}^N \in [0,1]^N$
**Output:** Reconstructed output $z^N \in \{0,1\}^N$
 1: $\Delta^N \leftarrow \text{DECOMPRESS}(b^*, \overline{P}^N)$
 2: **for** $i = 1, \ldots, N$ **do**
 3:      $P_i \leftarrow \Pr(U_i = 0 \mid U^{i-1} = u^{i-1}, \Pi)$
 4:      $\tilde{u}_i = \mathbf{1}\{s_i > P_i\}$
 5:      $u_i \leftarrow \Delta_i \oplus \tilde{u}_i$
 6: **end for**
 7: $z^N \leftarrow \text{INVERSEPOLARENC}(u^N, \Pi)$
 8: **return** $z^N$

---

simple scheme (from Sec. 3.1) to simulate these polarized subchannels.

Algorithms 1 and 2 describe the full scheme, retaining a similar structure to PolarSim where both the noise-free and the noisy subchannels are handled in a unified manner. The Compress and Decompress routines can be any prefix-free lossless entropy coding scheme. We use arithmetic coding (Rissanen, 1976) with a probability table that is computed offline via Monte Carlo estimation of $\Pr(\Delta_i = 1)$ and shared between the encoder and decoder.

The source-conditioned subchannel probabilities $\Pr(U_i = 0 \mid U^{i-1} = u^{i-1}, V^N = v^N, \Pi)$ for $i \in \{1, \ldots, N\}$ are computed for the given $n^*$ using a permutation-aware variant of Arıkan's recursive polar decoding algorithm (Arıkan, 2009). The required modification is straightforward, consisting only of accounting for the prescribed permutations at each recursion level, and therefore omitted for brevity.

The corresponding source-free conditionals $\Pr(U_i = 0 \mid U^{i-1} = u^{i-1})$ can also be evaluated using the same polar decoder for the degenerate channels $W_{Z_i|V}(Z_i \mid V = x) = P_{Z_i}(Z_i)$ for all $x \in \mathcal{V}$.

The InversePolarEnc routine computes the inverse of PermutedPolarTransform and recovers the simulation output by applying the same recursive transform in reverse order, using the inverse permutation at each recursion level

We show that our scheme is asymptotically optimal, achieving the mutual information lower bound in the large-block length limit.

**Theorem 3.1.** *Algorithms 1 and 2 satisfy the following:*

1. *They simulate the target ensemble of channels exactly.*

2. *There exists a choice for the probability table $\overline{P}^N$ such that the scheme is first-order optimal in its rate:*

$$\lim_{N \to \infty} \frac{E\left[\ell(b)\right] - \sum_{i=1}^N I(V_i; Z_i)}{N} = 0, \quad (10)$$

*where $\ell(\cdot)$ represents the length of a binary string (recall from Sec. 2.2).*

In practice, estimation errors in the probability table and the marginals $P_{Z_i}$ contribute to a slight overhead (even

considering large block lengths).

The overall complexity of the scheme matches that of `PolarSim`, which is $O(N \log N)$. Although random permutations are applied, they can be implemented with $O(N)$ time complexity per level (Knuth, 1981), ensuring an $O(N \log N)$ cumulative permutation overhead.

## 4. Neural compression with SoftBinary Coding

SBC uses the generalized `PolarSim` scheme to simulate independent realizations of the *softbinary* channel,

$$q_\theta(\cdot \mid x) = \text{Bern}\left(\frac{1+v}{2}\right), \text{ for } v = f_\theta(x) \in (-1, 1),$$

drawing latents $Z_i \sim q_\theta(\cdot \mid X_i)$. In this section, we discuss how the corresponding objective (1) can effectively be minimized using gradient-based optimization, in spite of the apparent discreteness. Further details are provided in Appendix D.

### 4.1. Training

As sampling from $q_\theta(Z \mid X)$ produces discrete values, it is a challenge to obtain a gradient for $\theta$ useful for stochastic optimization with the usual Monte Carlo method. However, in contrast to the scalar quantization of NTC, $q_\theta$ here is stochastic, which widens the choice of available gradient estimators. Here, we use the recently proposed unbiased and low-variance estimator *VarGrad* (Richter et al., 2020), which was developed for latent variable models such as VAEs. Central to VarGrad is the identity

$$\nabla_\theta D_{\text{KL}}\left(q_\theta(Z) \parallel p(Z \mid X)\right)$$
$$= \frac{1}{2}\nabla_\theta \text{Var}_{Z \sim r}\left(\log \frac{q_\theta(Z)}{p(Z \mid X)}\right)\Bigg|_{r=q_\theta}, \quad (11)$$

where $p(Z \mid X)$ is the (intractable) posterior. In words, the gradient of the Kullback–Leibler divergence (KLD) with respect to the parameters of the encoder $q_\theta$ can be obtained by evaluating the derivative of the variance of the log probability ratio, taken over samples from $q_\theta$—that is, as long as $q_\theta$ is stochastic, since the variance would otherwise be zero. Fortunately, it turns out that for SBC, it is, and that the proof for (11) given by Richter et al. (2020, Appendix A.1) holds for any differentiable expression in place of $p(Z \mid X)$. This makes it possible to apply the identity to the entire loss function (1) (although it may not represent a valid KLD) and obtain unbiased derivatives for $\theta$.

Even with unbiased gradient estimates, stochastic optimization is not guaranteed to find a global minimum of the loss function. While it is difficult to assess the characteristics of the loss function precisely, we have found that the same

optimization with a different initialization of model parameters $(\theta, \phi, \psi)$ can indeed yield slightly different results. To mitigate this, we used two techniques to reduce initialization dependence:

• We repeat each optimization with three different random initializations and select the one with the lowest loss.

• We add a regularization term $\|f_\theta(X)\|^2$ to the training loss, with a weight that is scheduled to decay exponentially from 0.5 at the beginning to negligible values around the middle of training. This keeps the encoder close to the decision boundary at the beginning of training, leading to a kind of annealing procedure.

We have found that these measures lead to consistent operational rate–distortion performance across experiments.

### 4.2. Operational scheme

The encoder $f_\theta$ maps a single source realization to $L$ channel inputs, where $L$ is a small power of two. While `PolarSim` can simulate channels of size $L$, its performance improves with increasing block length. Accordingly, we apply $f_\theta$ to $N/L$ independent realizations of the source, where $N$ is a large power of two, and apply `PolarSim` to the concatenated outputs, resulting in an effective block length of $N$. This is facilitated by the favorable scaling complexity of `PolarSim`. Our overall scheme thus resembles *concatenated codes* in communications (Forney, 1965). It is also analogous to how arithmetic coding is deployed in practice, where quantized elements from multiple spatial locations in an image are concatenated to approach the entropy limit. We found that performance is improved if latent bits that have a very small contribution to the KLD (identified by Monte Carlo estimating their KLD using source samples and thresholding it) are removed prior to concatenation.

The operational scheme is illustrated in Fig. 2.

## 5. Experiments

We evaluate SBC across a suite of information-theoretic sources. A key advantage of these benchmarks is that we can compare to analytical bounds from the literature, allowing us to investigate how the methods perform with respect to the theoretical optimum. In our experiments, we set the number of binary-output channels simulated per `PolarSim` block $N$ to be $N = 2^{23}$ ($\approx 8.4 \times 10^6$), allowing SBC to achieve polarization gains. Since each source realization is mapped to a latent dimension of at most $L$ (subject to the pruning process, in Sec. 4.2), the number of source realizations encoded by the $N$ bits is approximately $N/L$. The number of source samples processed by the encoder $f_\theta$ per call remains 1 for both NTC and SBC.

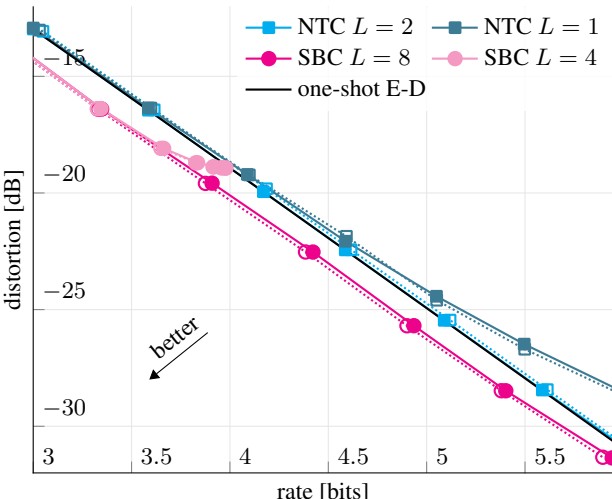

*Figure 3.* Rate–distortion performance on the circle. One-shot entropy–distortion (E-D) bound is due to Bhadane et al. (2022).

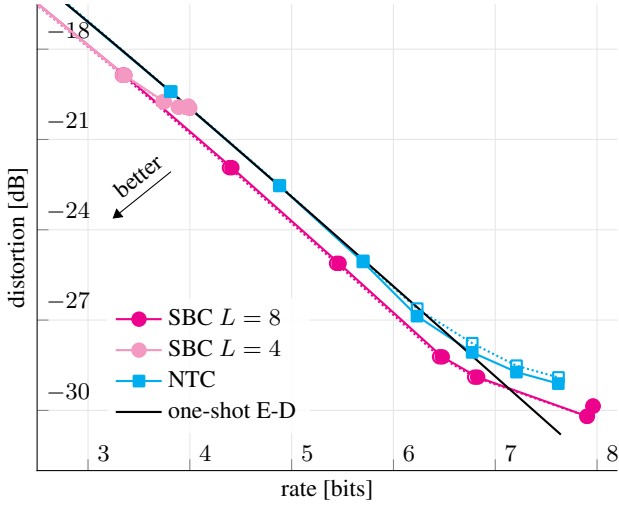

*Figure 4.* Rate–distortion performance on the ramp. One-shot entropy–distortion (E-D) bound is due to Bhadane et al. (2022).

It is worth highlighting our use of a larger latent dimension $L$ for SBC (4, 8, or 32 across experiments) compared to NTC ($L = 1$ in most cases). This follows directly from the two schemes' capacity constraints: each SBC latent dimension corresponds to a single bit, capping the rate at $L$ bits per source realization, whereas each NTC dimension is a real number. NTC can therefore realize a given rate with fewer dimensions, though its reliance on continuous-valued latents carries a smoothness bias (Bhadane et al., 2022; Ozyilkan et al., 2024a) that SBC avoids.

For NTC results, we opt for the annealing procedure by Agustsson & Theis (2020), which interpolates between soft and hard quantization to minimize the mismatch between training and operational losses, and we use the factorized entropy model in (Ballé et al., 2018, Appendix 6.1) for $p_\psi$.

In Fig. 3–8, the dotted and solid lines refer to the training and operational performances, respectively, both for NTC and SBC. In all cases, we use mean squared error as distortion.

Both **circle** and **ramp** sources were introduced by Bhadane et al. (2022) to highlight failure modes of NTC related to *smoothness bias* in the encoder and decoder transforms, respectively. The circle is a 2D source with an intrinsic dimensionality of one, with $X = (\cos\theta, \sin\theta) \in \mathbb{R}^2$, and $\theta \sim \mathcal{U}(0, 2\pi)$. Bhadane et al. (2022) showed that while a latent dimension of $L = 1$ should be sufficient to compress this source, NTC struggles to resolve the branch cut needed to represent it in its latent space. With $L = 2$, the issue can be worked around in this case, with one dimension serving as an indicator function and the other representing each half-circle (although this may not be possible for more complex sources with a circular topology). Fig. 3 shows that SBC is not subject to this smoothness issue, and also appears to obtain a shaping gain over the one-shot bound, as in the other examples to follow. Note that SBC is limited in another way—it can only operate up to a rate of $L$, as evident in the curve showing $L = 4$.

The **ramp** is a process defined as $X_t = [(t+\beta) \pmod 1] - \frac{1}{2}$, with latent phase $\beta \sim \mathcal{U}(0, 1)$. In this case, the source exhibits a sharp jump at a random point between 0 and 1. Fig. 4 shows that NTC achieves the one-shot entropy–distortion bound; however, at higher rates, it fails to recover the sharp discontinuities needed in the decoder. Again, SBC is not limited by this, but it is again by dimensionality: with $L = 4$ and $L = 8$, the representation is limited to rates of 4 and 8 bits, respectively. This is in contrast to NTC, where latent space dimensionality does not limit the rate.

**Distributed compression** involves the compression of multiple correlated sources that are encoded independently but decoded jointly. Here, we focus on the asymmetric case, first characterized by Wyner & Ziv (1976), which captures the fundamental challenge of exploiting source correlations when side information $Y$ is available exclusively at the decoder. The achievability proof in Wyner & Ziv (1976) for this setting posits a strategy known as "binning," which involves assigning source sequences into bins (subsets) and transmitting only the bin index instead of the sequence index. While this bin index is inherently ambiguous, the ambiguity can be resolved at the decoder with the help of side information, thereby yielding rate reduction.

This problem has been extensively studied; for practical schemes, the seminal work in the pre-ML era is DISCUS (Pradhan & Ramchandran, 2003) which implements binning mechanisms by exploiting ideas from channel coding (i.e., coset codes). More recently, Ozyilkan et al. (2023; 2024b) show that VQ-like parameterized neural networks recover behavior akin to binning, whereas ECSQ-style NTC cannot. This is, again, because binning as a quantization strategy

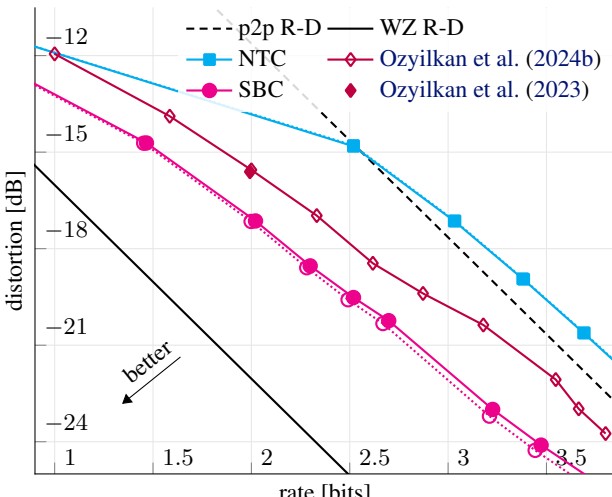

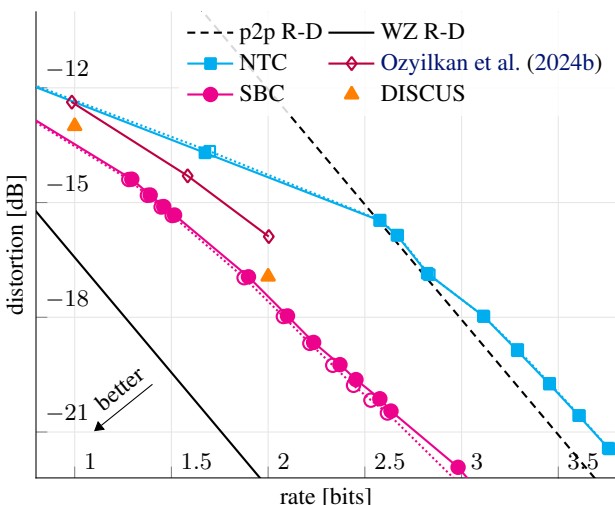

*Figure 5.* Rate–distortion performance on distributed compression of $X = Y + N$ with side information $Y \sim \mathcal{N}(0, 1)$ and $N \sim \mathcal{N}(0, 10^{-1})$ (NTC: $L = 1$; SBC: $L = 32$). The asymptotic rate–distortion bound (R-D) with side information, due to Wyner & Ziv (1976), is denoted as *WZ R-D*, and the asymptotic R-D bound without any side information (*point-to-point*), hence worse, is referred to as *p2p R-D*.

*Figure 6.* Rate–distortion performance on distributed compression of $Y = X + N$ with $X \sim \mathcal{N}(0, 1)$ and $N \sim \mathcal{N}(0, 10^{-1})$, where $Y$ is side information (NTC: $L = 1$; SBC: $L = 32$). For DIS-CUS by Pradhan & Ramchandran (2003), we include data points obtained with trellis-based quantization and coset construction, available at $R \in \{1, 2\}$ bits. The asymptotic rate–distortion bound (R-D) with side information, due to Wyner & Ziv (1976), is denoted as *WZ R-D*, and the asymptotic R-D bound without any side information (*point-to-point*), hence worse, is referred to as *p2p R-D*.

requires highly discontinuous encoder transforms $f_\theta$ (e.g., Ozyilkan et al., 2023, Fig. 2) in the case of NTC. Figs. 5 and 6 reveal that SBC again outperforms both the classical fixed-rate method DISCUS (which, notably, is constructed specifically for this source) as well as the more recent learning-based work, which like SBC is data-driven and does not require closed-form knowledge of the source.

The **i.i.d. Gaussian** source, where $X \sim \mathcal{N}(0, 1)$, is a well-studied setup in quantization theory with an optimal rate–distortion tradeoff given by $D(R) = 2^{-2R}$. Despite its simplicity, achieving performance close to this asymptotic bound remains a significant challenge. Trellis Coded Quantization (TCQ) (Marcellin & Fischer, 1990; Taubman & Marcellin, 2013) represents a low-complexity approach to implementing VQ, and has been the de facto standard for decades (Forney, 1992; Li et al., 2020). Lei et al. (2025) propose replacing the rectangular quantization grid ($\mathbb{Z}^L$) in NTC with another form of VQ: higher-dimensional lattices such as the $\Lambda_{24}$ Leech lattice (Leech, 1967; Conway & Sloane, 1999). SBC outperforms the state-of-the-art (TCQ) as well as the recent lattice-based solution (Fig. 7). Remarkably, it does so without explicit vector quantization. On the other hand, NTC is limited by its use of a scalar quantizer and hence performs as expected for an ECSQ.

The **i.i.d. uniform** source with $X \sim \mathcal{U}(-\frac{1}{2}, \frac{1}{2})$ is another canonical benchmark in the quantization literature. Fig. 8 reveals a specific failure mode in NTC: the smoothness bias of the entropy model. While the encoder transform $f_\theta$ only needs to implement the identity transformation, $p_\psi$ as a

continuous density function struggles to model the sharp boundaries of the uniform distribution, leading to model mismatch, and in turn to a suboptimal rate. SBC reproduces the R-D performance of TCQ, which is consistent with the known optimal points. In contrast to the fixed-rate method TCQ, SBC is also able to obtain intermediate points.

## 6. Discussion

SBC is a novel neural compression method which avoids hard quantization, and hence the disconnect between the training and operational R-D performance encountered in NTC. It achieves state-of-the-art compression performance on several information-theoretic sources. The *circle*, *ramp*, and *uniform* sources serve to demonstrate an inherent benefit over NTC, in terms of discontinuities in the encoder, the decoder, and the prior, respectively. With *binning* as a particularly challenging form of discontinuity, the distributed compression problem corroborates these findings. On top of this, SBC obtains shaping gains on all the reported sources—matching the state-of-the-art on the uniform source and setting a new one on the i.i.d. Gaussian source.

Due to SBC's rate limitation to $L$ bits per source sample, where $L$ is the number of latent dimensions (Sec. 5), resource requirements for training SBC on higher-dimensional sources grow faster than for NTC, similar to the so-called "curse of dimensionality" observed in fitting vector quan-

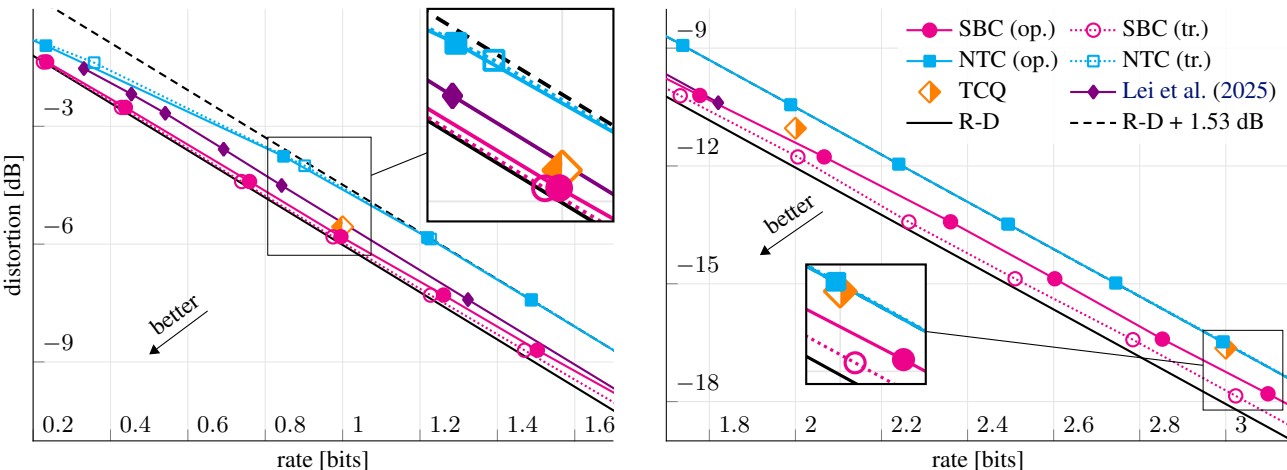

*Figure 7.* Rate–distortion performance on i.i.d. Gaussian source (NTC: $L = 1$; SBC: $L = 32$). The 1.53 dB offset accounts for the space-filling loss that the entropy-constrained scalar quantizer (ECSQ) is subjected to in a high-rate regime (Zamir, 2014). TCQ points with 256 states are obtained from Taubman & Marcellin (2013). We include the best performing ECLQ scheme by Lei et al. (2025), which uses $\Lambda_{24}$ Leech lattice.

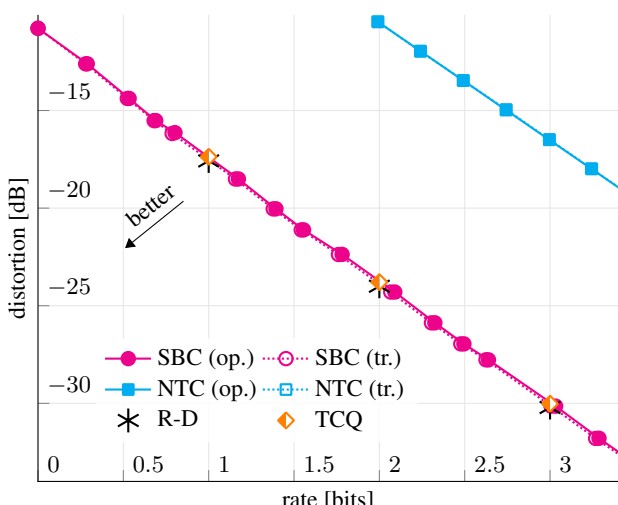

*Figure 8.* Rate–distortion performance on i.i.d. uniform source (NTC: $L = 1$; SBC: $L = 32$). Rate–distortion (R-D) points at $R = \{1, 2, 3\}$ bits and TCQ points with 256 states are obtained from Taubman & Marcellin (2013).

tizers. This currently limits our experiments to low-dimensional sources. However, we believe that in particular for the kind of low-rate applications that NTC is now increasingly being used for, such as point-cloud attributes, SBC could be a strong alternative, due to its better performance and weaker assumptions on source characteristics. Still, we plan to explore training SBC on high-dimensional sources such as images in future work. We also aim to explore how the efficiency of `PolarSim` can be increased at moderate block lengths by leveraging ideas from polar coding (e.g., Tal & Vardy, 2015), so that the large $N$ we use in our experiments can be reduced without incurring a rate penalty compared to the training rate.

A related training-time observation is that SBC exhibits mild initialization dependence, which we mitigate using the techniques in Sec. 4. We have not identified its precise cause, but the phenomenon is not unexpected: latent-variable models with discrete representations—VQ-VAE (van den Oord et al., 2017), Gumbel-Softmax / Concrete relaxations (Jang et al., 2017; Maddison et al., 2017), and even classical Lloyd–Max vector quantizers (Lloyd, 1982)—all exhibit similar behavior. NTC, by contrast, appears to escape this, plausibly because its training loss acts as a continuous proxy for an inherently discrete problem. A natural direction for future work is to develop an analogous coding scheme for *continuous* latents, possibly also based on polarization. Beyond broadening SBC's applicability, this would also let us test whether the initialization dependence is genuinely a consequence of the discrete latent space.

An intriguing aspect of SBC is that it obtains significant gains without using sophisticated probabilistic models. In practice, *hyperprior* (Ballé et al., 2018) or autoregressive entropy models (Minnen et al., 2018) require computing probabilities for arithmetic coding dynamically. This can be a significant implementational burden, as arithmetic coding requires bit-exact matching of probabilities on encoder and decoder sides, which can lead to catastrophic decoding errors when different floating-point hardware is involved (Ballé et al., 2019; Shi et al., 2024; Pang et al., 2024). With its simple factorized Bernoulli model, SBC appears to delegate the "heavy lifting" in terms of probabilistic modeling to the channel simulation part. If this holds true for complicated sources that cannot be sufficiently decorrelated using the encoder transform alone—another topic of future investigation—then SBC might have significant practical benefits in terms of implementational simplicity.

## Impact Statement

This paper explores theoretical and empirical ideas to advance the field of neural data compression. The broader societal consequences of advancing these learning-based compression methods are indirect and difficult to predict, none of which we feel must be specifically highlighted here.

## Acknowledgment

The AI language models ChatGPT (OpenAI, 2026) and Claude (Anthropic, 2026) were used to assist with editing the manuscript and debugging the source code. Certain arguments in the proof of Theorem B.1 were also explored in dialogue with ChatGPT (OpenAI, 2026).

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

## A. Randomized polar encoder

---

**Algorithm 3** `PermutedPolarTransform`

---

**Input:** Block length $N \in 2^{\mathbb{N}}$
**Input:** Bit sequence $z^N \in \{0, 1\}^N$
**Input:** Permutation randomness $\Pi \in \mathcal{S}$
**Input:** Polarization level $n^*$
**Output:** Output $u^N \in \{0, 1\}^N$
 1: **procedure** TRANSFORMSTEP($z^k, \Pi, m$)
 2: **if** $m = n^*$ **then**
 3:     **return** $z^k$
 4: **end if**
 5: $(\Pi_0, \Pi_1, \Pi_2) \leftarrow$ SPLIT($\Pi$)
 6: $\tilde{y}^k \leftarrow$ RANDOMPERMUTE($z^k, \Pi_0$)
 7: $\tilde{y}^{k/2}_{\text{even}} \leftarrow (\tilde{y}_{2i})^{k/2}_{i=1}$
 8: $\tilde{y}^{k/2}_{\text{odd}} \leftarrow (\tilde{y}_{2i-1})^{k/2}_{i=1}$
 9: $u^{k/2} \leftarrow$ TRANSFORMSTEP($\tilde{y}^{k/2}_{\text{even}} \oplus \tilde{y}^{k/2}_{\text{odd}}, \Pi_1, m+1$)
10: $\tilde{u}^{k/2} \leftarrow$ TRANSFORMSTEP($\tilde{y}^{k/2}_{\text{even}}, \Pi_2, m+1$)
11: **return** $(u^{k/2}, \tilde{u}^{k/2})$
12: **end procedure**
13: **return** TRANSFORMSTEP($z^N, \Pi, 0$)

---

Here, the function RANDOMPERMUTE($\cdot, \Pi$) applies a uniformly random permutation to the input sequence, with the randomness contingent on the seed $\Pi$. The seed-splitting routine SPLIT($\cdot$) produces independent sub-seeds, ensuring that each permutation is generated independently of the others. The parameter $n^* \in \{0, \dots, n\}$ indicates the number of recursive levels we use. There is a technical condition that arises in Theorem B.1 which necessitates the inclusion of the $n^*$ parameter. In our experiments, we fix $n^* = n$.

Random permutations are applied to symmetrize the subchannels at every recursion level, facilitating the analysis of the scheme. In practice, we obtain satisfactory empirical results even without these permutations.

## B. Nonstationary source polarization theorem

**Theorem B.1.** *Let $N = 2^n$, and let $\Pi$ be an independent random permutation, drawn from the common randomness, that determines the random permutations used by* `PermutedPolarTransform`*. Then, for every $\delta \in (0, 1)$ and $n$, there exists $n^* \in \{1, \dots, n\}$ such that, with*

$$U^N = \texttt{PermutedPolarTransform}\left(Z^N, \Pi, n^*\right), \tag{12}$$

*we have*

$$\lim_{n \to \infty} \frac{\left|\left\{i \in \{1, \dots, 2^n\} : H(U_i \mid U^{i-1}, V^N, \Pi) \in (\delta, 1 - \delta)\right\}\right|}{2^n} = 0. \tag{13}$$

### B.1. Proof of Theorem B.1

We will begin by defining a measure of channel quality called the *Bhattacharyya parameter*.

**Definition B.2** (cf. (Arıkan, 2010))**.** Consider a pair of random variables $T, Z \sim P_{TZ}$ taking values in any $\mathcal{T} \times \{0, 1\}$. We then define

$$\mathcal{B}(P_{Z|T}) = 2E_T\left[\sqrt{P_{Z|T}(0 \mid T)P_{Z|T}(1 \mid T)}\right]. \tag{14}$$

The Bhattacharyya parameter contracts under a single step of the `PermutedPolarTransform`.

**Definition B.3** (Pairwise polarizing transform)**.** Let $(T_1, Z_1)$ and $(T_2, Z_2)$ be independent random pairs with joint distributions $P_{T_1 Z_1}$ and $P_{T_2 Z_2}$, respectively, with $Z_1$ and $Z_2$ being binary valued.

Define the channels

$$W_1(z_1 \mid t_1) := P_{Z_1 \mid T_1}(z_1 \mid t_1), \tag{15}$$

$$W_2(z_2 \mid t_2) := P_{Z_2 \mid T_2}(z_2 \mid t_2). \tag{16}$$

From $(W_1, W_2)$, define the transformed channels

$$\check{W}(z \mid t_1, t_2) = \Pr(Z_1 \oplus Z_2 = y \mid T_1 = t_1, T_2 = t_2), \tag{17}$$

$$\hat{W}(z_2 \mid t_1, t_2, z) = \Pr(Z_2 = z_2 \mid T_1 = t_1, T_2 = t_2, Z_1 \oplus Z_2 = z). \tag{18}$$

We write

$$(W_1, W_2) \mapsto (\check{W}, \hat{W}) \tag{19}$$

to denote this channel transformation, and refer to it as the *pairwise polarizing transform*.

**Lemma B.4** (cf. Proposition 1, (Arıkan, 2010))**.** *Under a pairwise polarizing transform, we have*

$$\mathcal{B}(\hat{W}) = \mathcal{B}(W_1) \cdot \mathcal{B}(W_2), \text{ and} \tag{20}$$

$$\mathcal{B}(\check{W}) + \mathcal{B}(\hat{W}) \leq \mathcal{B}(W_1) + \mathcal{B}(W_2). \tag{21}$$

*Proof.* For $z_2 \in \{0, 1\}$, we have

$$\Pr(Z_2 = z_2 \mid T^2 = t^2, Z_1 \oplus Z_2 = z) \tag{22}$$

$$= \frac{\Pr(Z_2 = z_2, Z_1 \oplus Z_2 = z \mid T^2 = t^2)}{\Pr(Z_1 \oplus Z_2 = z \mid T^2 = t^2)} \tag{23}$$

$$= \frac{\Pr(Z_2 = z_2, Z_1 = z \oplus z_2 \mid T^2 = t^2)}{\Pr(Z_1 \oplus Z_2 = z \mid T^2 = t^2)} \tag{24}$$

$$= \frac{\Pr(Z_2 = z_2 \mid T_2 = t_2) \Pr(Z_1 = z \oplus z_2 \mid T_1 = t_1)}{\Pr(Z_1 \oplus Z_2 = z \mid T^2 = t^2)}. \tag{25}$$

Therefore,

$$\mathcal{B}(\hat{W}) \tag{26}$$

$$= 2 E_{T^2, Z} \left[ \sqrt{\frac{\Pr(Z_2 = 0 \mid T_2) \Pr(Z_1 = Z \oplus 0 \mid T_1) \Pr(Z_2 = 1 \mid T_2) \Pr(Z_1 = Z \oplus 1 \mid T_1)}{(\Pr(Z_1 \oplus Z_2 = Z \mid T^2))^2}} \right] \tag{27}$$

$$= 4 E_{T^2} \left[ \sqrt{\Pr(Z_2 = 0 \mid T_2) \Pr(Z_1 = 0 \mid T_1) \Pr(Z_2 = 1 \mid T_2) \Pr(Z_1 = 1 \mid T_1)} \right] \tag{28}$$

$$= 2 E_{T_1} \left[ \sqrt{\Pr(Z_1 = 1 \mid T_1) \Pr(Z_1 = 0 \mid T_1)} \right] \cdot 2 E_{T_2} \left[ \sqrt{\Pr(Z_2 = 1 \mid T_2) \Pr(Z_2 = 0 \mid T_2)} \right] \tag{29}$$

$$= \mathcal{B}(W_1) \cdot \mathcal{B}(W_2). \tag{30}$$

Similarly,

$$\Pr(Z_1 \oplus Z_2 = z \mid T^2 = v^2) \tag{31}$$

$$= \sum_{z_2 \in \{0,1\}} \Pr(Z_1 \oplus Z_2 = z, Z_2 = z_2 \mid T^2 = v^2) \tag{32}$$

$$= \sum_{z_2 \in \{0,1\}} \Pr(Z_1 = z \oplus z_2 \mid T_1) \Pr(Z_2 = z_2 \mid T_2 = T_2). \tag{33}$$

Therefore

$$\mathcal{B}(\check{W}) \tag{34}$$

$$= 2E_{T^2}\left[\sqrt{\Pr(Z_1 = 0 \mid T_1)\Pr(Z_2 = 0 \mid T_2) + \Pr(Z_1 = 1 \mid T_1)\Pr(Z_2 = 1 \mid T_2)} \right. \tag{35}$$

$$= \left. \phantom{2E_{T^2}} \cdot \sqrt{\Pr(Z_1 = 1 \mid T_1)\Pr(Z_2 = 0 \mid T_2) + \Pr(Z_1 = 0 \mid T_1)\Pr(Z_2 = 1 \mid T_2)}\right]. \tag{36}$$

Now, for $\alpha, \beta \in (0, 1)$, consider

$$\sqrt{(\alpha\beta + (1 - \alpha)(1 - \beta)) \cdot (\alpha(1 - \beta) + \beta(1 - \alpha))} \tag{37}$$

$$= \sqrt{(\alpha\beta + (1 - \alpha)(1 - \beta)) \cdot (\alpha(1 - \beta) + \beta(1 - \alpha))} \tag{38}$$

$$= \sqrt{\alpha(1 - \alpha) + \beta(1 - \beta) - 4\alpha\beta(1 - \alpha)(1 - \beta)} \tag{39}$$

Set $p = \sqrt{\alpha(1 - \alpha)}$ and $q = \sqrt{\beta(1 - \beta)}$. Then, we have

$$\alpha(1 - \alpha) + \beta(1 - \beta) - 4\alpha\beta(1 - \alpha)(1 - \beta) \tag{40}$$

$$- \left(\sqrt{\alpha(1 - \alpha)} + \sqrt{\beta(1 - \beta)} - 2\sqrt{\alpha(1 - \alpha)\beta(1 - \beta)}\right)^2 \tag{41}$$

$$= p^2 + q^2 - 4p^2q^2 - (p + q - 2pq)^2 \tag{42}$$

$$= 4pq^2 + 4qp^2 - 8p^2q^2 - 2qp \tag{43}$$

$$= -2pq(2p - 1)(2q - 1) \le 0. \tag{44}$$

Here (44) follows from the fact that $p \le \frac{1}{2}$ and $q \le \frac{1}{2}$. Substituting this result in (36), and combining with (30), we obtain the required result. □

Our goal is to use this contraction to show that the Bhattacharyya parameters of the subchannels are polarized: $\mathcal{B}(U_i \mid U^{i-1}, V^N, \Pi) \approx 0$ or $\mathcal{B}(U_i \mid U^{i-1}, V^N, \Pi) \approx 1$ for most indices. Our proof strategy will be to construct a Lyapunov-like function of the subchannel Bhattacharyya parameters that decreases strictly through each pairwise transform step for non-polarized channels.

Fix $\gamma \in (0, 1)$ and define $\phi(x) = x^\gamma$, for $x \in [0, 1]$.

**Lemma B.5.** *Consider the pairwise polarizing transform (see Definition B.3) applied to the pair $(W_1, W_2)$. Then, if $\mathcal{B}(W_1), \mathcal{B}(W_2) \in (\delta, 1 - \delta)$, there exists a positive constant $C(\delta, \gamma)$ s.t.*

$$\phi(\mathcal{B}(\hat{W})) + \phi(\mathcal{B}(\check{W})) \le \phi(\mathcal{B}(W_1)) + \phi(\mathcal{B}(W_2)) - C(\delta, \gamma). \tag{45}$$

*Proof.* Using (21) and the monotonicity of $\phi(\cdot)$, we have

$$\phi(\mathcal{B}(\hat{W})) + \phi(\mathcal{B}(\check{W})) \le \phi(\mathcal{B}(W_1) + \mathcal{B}(W_2) - \mathcal{B}(W_1)\mathcal{B}(W_2)) + \phi(\mathcal{B}(W_1)\mathcal{B}(W_2)). \tag{46}$$

Define

$$C(\delta, \gamma) = \min_{\delta \le a \le b \le 1 - \delta} f(a, b), \text{ where} \tag{47}$$

$$f(a, b) = \phi(a) + \phi(b) - \phi(a + b - ab) - \phi(ab). \tag{48}$$

Applying Lemma B.6 shows that $C(\delta, \gamma) > 0$. Combining this with (46) completes the proof. □

**Lemma B.6.** *For all $a, b \in (0, 1)$, the following holds:*

$$\phi(a + b - ab) + \phi(ab) < \phi(a) + \phi(b). \tag{49}$$

*Proof.* The proof follows by applying Karamata's inequality (Kadelburg et al., 2005), noting that $\phi(\cdot)$ is strictly concave and that $ab < a \le b < (a + b - ab)$ (where we assume w.l.o.g. that $a \le b$). □

We will begin by introducing some notation. Let $m \in \{0, 1, \ldots, n\}$ denote the recursion depth of the `PermutedPolarTransform`, where $m = 0$ denotes the initial sequence $Z^N$, and each increase in $m$ corresponds to one level of the recursive pairwise polarizing transform.

For a given depth $m \geq 1$, the recursion generates, for each binary string $B^m = (B_1, \ldots, B_m) \in \{0, 1\}^m$, a collection of synthetic subchannels associated with the branch $B^m$. Here, $B_i = 0$ indicates that, at level $i$ of the recursion, the subchannels associated with the $\check{W}$ output of the pairwise polarizing transform are followed, while $B_i = 1$ indicates that the subchannels associated with the $\hat{W}$ output are followed.

We will also let $\Pi^m = (\Pi_1, \ldots, \Pi_m)$ denote the permutation randomness at each stage (see Algorithm 3), with each $\Pi_i$ corresponding to the random permutation applied to the sequence at stage $i - 1$.

For fixed $(B^m, \Pi^m, n)$, denote the corresponding subchannels by

$$W_1(B^m, \Pi^m, n), \ldots, W_{N_m}(B^m, \Pi^m, n), \text{ where } N_m = 2^{n-m}$$

is the number of subchannels at this level. Then, define

$$\Phi(B^m, \Pi^m, n) = \frac{1}{N_m} \sum_{j=1}^{N_m} \phi(\mathcal{B}(W_j(B^m, \Pi^m, n))). \tag{50}$$

At the next $m + 1$ stage of polarization, the current subchannels are randomly paired using $\Pi_{m+1}$, generating $N_m/2$ pairs

$$(W_i^a(B^m, \Pi^{m+1}), W_i^b(B^m, \Pi^{m+1}))_{i=1}^{\frac{N_m}{2}}, \text{ s.t.} \tag{51}$$
$$\mathcal{B}(W_i^a(B^m, \Pi^{m+1})) \leq \mathcal{B}(W_i^b(B^m, \Pi^{m+1})) \text{ for all } i, \tag{52}$$

which are then combined using the pairwise polarizing transform

$$(W_i^a(B^m, \Pi^{m+1}), W_i^b(B^m, \Pi^{m+1})) \mapsto (\check{W}_i(B_0^{m+1}, \Pi^{m+1}), \hat{W}_i(B_1^{m+1}, \Pi^{m+1})), \tag{53}$$

where $B_b^{m+1} = (B_1, \ldots, B_m, b)$. We then have

$$E_{\Pi_{m+1}, B_{m+1}} \left[ \Phi(B^{m+1}, \Pi^{m+1}, n) \mid B^m, \Pi^m \right] \tag{54}$$

$$= \frac{1}{N_m} \sum_{i=1}^{N_m/2} E_{\Pi_{m+1}} \left[ \phi(\mathcal{B}(\check{W}_i(B_0^{m+1}, \Pi^{m+1}))) + \phi(\mathcal{B}(\hat{W}_i(B_1^{m+1}, \Pi^{m+1}))) \right], \tag{55}$$

with the expectation being taken over $B_{m+1} \sim \text{Bern}\left(\frac{1}{2}\right)$ and the permutation randomness $\Pi_{m+1}$. From Lemma B.5, we then obtain

$$E_{\Pi_{m+1}, B_{m+1}} \left[ \Phi(B^{m+1}, \Pi^{m+1}, n) \mid B^m, \Pi^m \right] \tag{56}$$
$$\leq \Phi(B^m, \Pi^m, n)$$

$$- C(\delta, \gamma) E_{\Pi_{m+1}} \left[ \frac{1}{N_m} \sum_{i=1}^{N_m/2} \mathbf{1} \left\{ \delta \leq \mathcal{B}(W_i^a(B^m, \Pi^{m+1}, n)) \leq \mathcal{B}(W_i^b(B^m, \Pi^{m+1}, n)) \leq 1 - \delta \right\} \right] \tag{57}$$

$$= \Phi(B^m, \Pi^m, n)$$
$$- \frac{C(\delta, \gamma)}{2} P_{\Pi_{m+1}} \left( \delta \leq \mathcal{B}(W_i^a(B^m, \Pi^{m+1}, n)) \leq \mathcal{B}(W_i^b(B^m, \Pi^{m+1}, n)) \leq 1 - \delta \right). \tag{58}$$

Now, consider the fraction of indices $j$ which are non-polarized at level $m$:

$$\theta(B^m, \Pi^m, n) = \frac{|\{1 \leq j \leq N_m : \mathcal{B}(W_j(B^m, \Pi^m, n)) \in (\delta, 1 - \delta)\}|}{N_m}. \tag{59}$$

Then, we have

$$P_{\Pi_{m+1}}\left(\delta \leq \mathcal{B}(W_i^a(B^m, \Pi^m, n)) \leq \mathcal{B}(W_i^b(B^m, \Pi^m, n)) \leq 1 - \delta\right) \tag{60}$$

$$= \theta(B^m, \Pi^m, n) \cdot \frac{(\theta(B^m, \Pi^m, n) \cdot N_m - 1)}{N_m - 1} \tag{61}$$

$$\geq \theta(B^m, \Pi^m, n)^2 - \frac{\theta(B^m, \Pi^m, n)}{N_m - 1}. \tag{62}$$

Substituting this in (58), we obtain

$$E_{\Pi_{m+1}, B_{m+1}}\left[\Phi(B^{m+1}, \Pi^{m+1}, n) \mid B^m, \Pi^m\right] \tag{63}$$

$$\leq \Phi(B^m, \Pi^m, n) - \frac{C(\delta, \gamma)}{2}\left(\theta(B^m, \Pi^m, n)^2 - \frac{\theta(B^m, \Pi^m, n)}{N_m - 1}\right). \tag{64}$$

Taking expectations over uniformly random bits $B_1, \ldots, B_m$ and permutations $\Pi_1, \ldots, \Pi_m$, and using Jensen's inequality to bound $E[\theta^2] \geq E[\theta]^2$, we then obtain

$$\overline{\Phi}_{m+1, n} \leq \overline{\Phi}_{m, n} - \frac{C(\delta, \gamma)}{2}\left(\overline{\theta}_{m, n}^2 - \frac{\overline{\theta}_{m, n}}{N_m - 1}\right), \tag{65}$$

where we have

$$\overline{\theta}_{m, n} = E_{B^m, \Pi^m}\left[\theta(B^m, \Pi^m, n)\right], \text{ and} \tag{66}$$

$$\overline{\Phi}_{m, n} = E_{B^m, \Pi^m}\left[\Phi(B^m, \Pi^m, n)\right]. \tag{67}$$

Using (65) to express the gap between $\overline{\Phi}_{n, n}$ and $\overline{\Phi}_{1, n}$ results in the following inequality:

$$\overline{\Phi}_{n, n} \leq \overline{\Phi}_{1, n} - \frac{C(\delta, \gamma)}{2} \sum_{i=1}^{n-1}\left(\overline{\theta}_{i, n}^2 - \frac{1}{2^i - 1}\right). \tag{68}$$

Upon rearranging, normalizing and taking the limit, we see that

$$\lim_{n \to \infty} \sup \frac{1}{n - 1} \sum_{i=1}^{n-1} \overline{\theta}_{i, n}^2 = 0. \tag{69}$$

Let us select

$$n^* = \min_{1 \leq j \leq n} \overline{\theta}_{j, n}. \tag{70}$$

Then,

$$\overline{\theta}_{n^*, n}^2 \leq \frac{1}{n} \sum_{i=1}^{n} \overline{\theta}_{i, n}^2 \tag{71}$$

$$\implies \lim_{n \to \infty} \sup \overline{\theta}_{n^*, n} = 0. \tag{72}$$

Bounding the subchannel conditional entropies in terms of the Bhattacharyya parameters (see Proposition 2, (Arıkan, 2010)) concludes the proof.

## C. Proof of Theorem 3.1

The correctness of the scheme follows from the fact that for a fixed permutation randomness, `PermutedPolarTransform` is a bijection.

To characterize the rate, for any $i \in \{1, \ldots, N\}$, consider

$$\Pr(\Delta_i = 1) \tag{73}$$

$$= E_{V^N, U^{i-1}, \Pi} \left[ \Pr(\Delta_i = 1 \mid V^N, U^{i-1}, \Pi) \right] \tag{74}$$

$$= E_{V^N, U^{i-1}, \Pi} \left[ \left| \Pr(U_i = 1 \mid V^N, U^{i-1}, \Pi) - \Pr(U_i = 1 \mid U^{i-1}, \Pi) \right| \right] \tag{75}$$

$$\leq \sqrt{E_{V^N, U^{i-1}, \Pi} \left[ (\Pr(U_i = 1 \mid V^N, U^{i-1}, \Pi) - \Pr(U_i = 1 \mid U^{i-1}, \Pi))^2 \right]} \tag{76}$$

$$= \sqrt{E_{U^{i-1}, \Pi} E_{V^N | U^{i-1}, \Pi} \left[ (\Pr(U_i = 1 \mid V^N, U^{i-1}, \Pi) - \Pr(U_i = 1 \mid U^{i-1}, \Pi))^2 \right]} \tag{77}$$

$$= \sqrt{E_{U^{i-1}, \Pi} E_{V^N | U^{i-1}, \Pi} \left[ (\Pr(U_i = 1 \mid V^N, U^{i-1}, \Pi) - E_{V^N | U^{i-1}, \Pi} \left[ \Pr(U_i = 1 \mid V^N, U^{i-1}, \Pi) \right])^2 \right]} \tag{78}$$

$$= \sqrt{E_{U^{i-1}, \Pi} \left[ \mathrm{Var} \left[ \Pr(U_i = 1 \mid V^N, U^{i-1}, \Pi) \mid U^{i-1}, \Pi \right] \right]} \tag{79}$$

$$= \sqrt{\mathrm{Var} \left[ \Pr(U_i = 1 \mid V^N, U^{i-1}, \Pi) \right] - \mathrm{Var} \left[ E \left[ \Pr(U_i = 1 \mid V^N, U^{i-1}, \Pi) \mid U^{i-1}, \Pi \right] \right]} \tag{80}$$

$$= \sqrt{\mathrm{Var} \left[ \Pr(U_i = 1 \mid V^N, U^{i-1}, \Pi) \right] - \mathrm{Var} \left[ \Pr(U_i = 1 \mid U^{i-1}, \Pi) \right]}. \tag{81}$$

Here (76) follows from the Cauchy-Schwarz inequality, and (80) follows from the law of total variance. Using Lemma C.1, we then obtain

$$\Pr(\Delta_i = 1) \leq \frac{1}{2} \sqrt{H(U_i \mid U^{i-1}, \Pi) - (H(U_i \mid U^{i-1}, V^N, \Pi))^2}. \tag{82}$$

We can then bound the rate of the scheme as follows:

$$\frac{1}{N} E[\ell(b)] = \frac{1}{N} \sum_{i=1}^N H(\Delta_i) + \frac{2}{N} \tag{83}$$

$$= \frac{1}{N} \sum_{i=1}^N h_B(\Pr(\Delta_i = 1)) + \frac{2}{N} \tag{84}$$

$$\leq \frac{1}{N} \sum_{i=1}^N h_B \left( \frac{1}{2} \sqrt{H(U_i \mid U^{i-1}, \Pi) - (H(U_i \mid U^{i-1}, V^N, \Pi))^2} \right) + \frac{2}{N}, \tag{85}$$

where $h_B(\cdot)$ is the binary entropy function.

Fix $\delta \in \left( 0, \frac{1}{2} \right)$. Next, we will partition the set of subchannel indices $i \in \{1, 2, \ldots, N\}$ into three sets:

$$A_{\mathrm{low}} = \{i : H(U_i \mid U^{i-1}, \Pi) \in (0, \delta)\}, \tag{86}$$

$$A_{\mathrm{mid}} = \{i : H(U_i \mid U^{i-1}, \Pi) \in (\delta, 1 - \delta)\}, \tag{87}$$

$$A_{\mathrm{high}} = \{i : H(U_i \mid U^{i-1}, \Pi) \in (1 - \delta, 1)\}. \tag{88}$$

Using these partitions, we can then write

$$\frac{1}{N} E[\ell(b)] \leq \frac{1}{N} \sum_{i=1}^N h_B \left( \frac{1}{2} \sqrt{H(U_i \mid U^{i-1}, \Pi) - (H(U_i \mid U^{i-1}, V^N, \Pi))^2} \right) + \frac{2}{N} \tag{89}$$

$$\leq \frac{1}{N} \sum_{i \in A_{\mathrm{high}}} h_B \left( \frac{1}{2} \sqrt{H(U_i \mid U^{i-1}, \Pi) - (H(U_i \mid U^{i-1}, V^N, \Pi))^2} \right)$$

$$+ h_B \left( \frac{\sqrt{\delta}}{2} \right) + \frac{|A_{\mathrm{mid}}|}{N} + \frac{2}{N}. \tag{90}$$

The bijectivity of the `PermutedPolarTransform` (upon fixing the permutations) implies that

$$\sum_{i=1}^{N} H(U_i \mid U_1^{i-1}, \Pi) = \sum_{i=1}^{n} H(Z_i). \tag{91}$$

Using Theorem B.1, we know that

$$\lim_{N \to \infty} \frac{|A_{\text{mid}}|}{N} = 0. \tag{92}$$

Hence, for $N$ sufficiently large s.t. $\frac{|A_{\text{mid}}|}{N} < \delta$,

$$(1 - \delta)|A_{\text{high}}| \leq \sum_{i=1}^{N} H(U_i \mid U_1^{i-1}, \Pi) \leq |A_{\text{high}}| + (1 - \delta)|A_{\text{mid}}| + \delta|A_{\text{low}}| \tag{93}$$

$$(1 - \delta)|A_{\text{high}}| \leq \sum_{i=1}^{N} H(Z_i) \leq |A_{\text{high}}| + (1 - \delta)|A_{\text{mid}}| + \delta|A_{\text{low}}| \tag{94}$$

$$(1 - \delta)\frac{|A_{\text{high}}|}{N} \leq \frac{1}{N} \sum_{i=1}^{N} H(Z_i) \leq \frac{|A_{\text{high}}|}{N} + 2\delta \tag{95}$$

$$-\delta \leq \frac{\sum\limits_{i=1}^{N} H(Z_i) - |A_{\text{high}}|}{N} \leq 2\delta. \tag{96}$$

Therefore,

$$\frac{|A_{\text{high}}|}{N} \leq \frac{1}{N} \sum_{i=1}^{N} H(Z_i) + 2\delta. \tag{97}$$

Using these bounds, and noting that $h_B\left(\frac{\sqrt{\delta}}{2}\right) \leq 2\delta^{1/4}$, we can express the rate as

$$
\begin{aligned}
\frac{1}{N} E[\ell(b)] &\leq \frac{1}{N} \sum_{i \in A_{\text{high}}} h_B\left(\frac{1}{2}\sqrt{H(U_i \mid U^{i-1}, \Pi) - (H(U_i \mid U^{i-1}, V^N, \Pi))^2}\right) \\
&\quad + \frac{1}{N} \sum_{i \in A_{\text{low}}} h_B\left(\frac{\sqrt{\delta}}{2}\right) + \frac{|A_{\text{mid}}|}{N} + \frac{2}{N} \\
&\leq \frac{1}{N} \sum_{i \in A_{\text{high}}} h_B\left(\frac{1}{2}\sqrt{H(U_i \mid U^{i-1}, \Pi) - (H(U_i \mid U^{i-1}, V^N, \Pi))^2}\right) \\
&\quad + 2\delta^{1/4} + \delta + \frac{2}{N}.
\end{aligned}
\tag{98}
$$
$$\tag{99}$$

To analyze the contribution of the high-entropy indices

$$\tilde{R} = \frac{1}{N} \sum_{i \in A_{\text{high}}} h_B\left(\frac{1}{2}\sqrt{H(U_i \mid U^{i-1}, \Pi) - (H(U_i \mid U^{i-1}, V^N, \Pi))^2}\right),$$

we consider another partition of the indices, based on the conditional entropies $H(U_i \mid U^{i-1}, V^N, \Pi)$:

$$B_{\text{low}} = \{i : H(U_i \mid U^{i-1}, V^N, \Pi) \in (0, \delta)\}, \tag{100}$$
$$B_{\text{mid}} = \{i : H(U_i \mid U^{i-1}, V^N, \Pi) \in (\delta, 1 - \delta)\}, \tag{101}$$
$$B_{\text{high}} = \{i : H(U_i \mid U^{i-1}, V^N, \Pi) \in (1 - \delta, 1)\}. \tag{102}$$

Then,

$$\tilde{R} = \frac{1}{N} \sum_{i \in A_{\text{high}}} h_B \left( \frac{1}{2} \sqrt{H(U_i \mid U^{i-1}, \Pi) - (H(U_i \mid U^{i-1}, V^N, \Pi))^2} \right) \tag{103}$$

$$\leq \frac{1}{N} \sum_{i \in A_{\text{high}} \cap B_{\text{low}}} h_B \left( \frac{1}{2} \sqrt{H(U_i \mid U^{i-1}, \Pi) - (H(U_i \mid U^{i-1}, V^N, \Pi))^2} \right)$$

$$+ \frac{|A_{\text{high}} \cap B_{\text{mid}}|}{N} + h_B \left( \sqrt{\frac{\delta}{2}} \right). \tag{104}$$

Here, (104) follows by observing that $h_B(\cdot) \leq 1$, and

$$H(U_i \mid U^{i-1}, \Pi) - (H(U_i \mid U^{i-1}, V^N, \Pi))^2 \tag{105}$$

$$\leq 1 - (1 - \delta)^2 \tag{106}$$

$$\leq 2\delta, \text{ for } i \in A_{\text{high}} \cap B_{\text{high}}. \tag{107}$$

Applying Theorem B.1, and performing similar computations to (93) - (96), we obtain

$$\frac{|B_{\text{mid}}|}{N} \leq 2\delta, \text{ and} \tag{108}$$

$$\frac{|B_{\text{high}}|}{N} \in \left( \frac{1}{N} \sum_{i=1}^{N} H(Z_i \mid V_i) - \delta, \frac{1}{N} \sum_{i=1}^{N} H(Z_i \mid V_i) + 2\delta \right). \tag{109}$$

From these, and noting that $B_{\text{high}} \subset A_{\text{high}}$, we can obtain an upper bound on $|A_{\text{high}} \cap B_{\text{low}}|$:

$$\frac{1}{N} |A_{\text{high}} \cap B_{\text{low}}| \leq \frac{1}{N} |A_{\text{high}}| - \frac{1}{N} |A_{\text{high}} \cap B_{\text{high}}| \tag{110}$$

$$= \frac{1}{N} |A_{\text{high}}| - \frac{1}{N} |B_{\text{high}}| \tag{111}$$

$$\leq \frac{1}{N} \sum_{i=1}^{N} I(V_i; Z_i) + 3\delta. \tag{112}$$

Substituting this in (104), and noting the bound $h_B \left( \sqrt{\frac{\delta}{2}} \right) \leq 2\delta^{1/4}$, we obtain

$$\tilde{R} \leq \frac{1}{N} \sum_{i=1}^{N} I(V_i; Z_i) + 4\delta + 2\delta^{1/4}. \tag{113}$$

Substituting this into (99), and considering $n > \frac{2}{\delta}$, we obtain

$$\frac{1}{N} E\left[ \ell(b) \right] \leq \frac{1}{N} \sum_{i=1}^{N} I(V_i; Z_i) + 6\delta + 4\delta^{1/4}. \tag{114}$$

The result then follows from this as $\delta$ was chosen arbitrarily.

**Lemma C.1.** *Given random variables $T, V \sim P_{T,V}$ on $\{0, 1\} \times \mathcal{V}$, define the quantities $P = \Pr(T = 1 \mid V)$, $p = \Pr(T = 1)$, and $H = H(T \mid V)$. We then have*

$$\text{Var}[P] - p(1 - p) \in \left( -\frac{H}{4}, -\frac{H^2}{4} \right). \tag{115}$$

*Proof.* First, we know that

$$\text{Var}[P] = E[P^2] - p^2 \tag{116}$$

$$= p(1 - p) - E[P(1 - P)]. \tag{117}$$

From standard bounds on the binary entropy function (see Theorem 1.2, (Topsøe, 2001)), we also have

$$H = E\left[h_B(P)\right] \tag{118}$$

$$\in \left(E\left[4P(1-P)\right], 2\sqrt{E\left[P(1-P)\right]}\right). \tag{119}$$

Combining (117) and (119), we obtain the required result. □

## D. More details on the training and operational schemes

Our experiments were designed to ensure a fair comparison between SoftBinary Coding (SBC) and the Nonlinear Transform Coding (NTC) baseline while maintaining high numerical stability. We equalized the sampling budget for both schemes by using a total of 256 samples for each optimization step. For SBC, we employed a batch size of 16, with 16 samples drawn from the encoder $q_\theta$ per source realization for the VarGrad objective, whereas NTC utilized a standard batch size of 256. We found that the optimization of SBC is robust with respect to allocating the total number of samples between samples from the encoder and from the source. To cover a sufficiently broad rate–distortion region of interest for both SBC and NTC, we varied the Lagrange multiplier $\lambda$ in (1) logarithmically.

Both models were optimized using the Adam optimizer for 10000 epochs (consisting of 1000 steps each) with a learning rate of $10^{-4}$, which was dropped to $10^{-5}$ during the final $10\%$ of training. Following standard practice in the neural data compression literature, we report discrete entropy estimates under the assumption that an ideal arithmetic coder would asymptotically reach this limit.

As discussed in Sec. 4.2, for the channel simulation part, we further refine coding efficiency by implementing a latent pruning strategy during the transition from the training to the operational scheme. While the end-to-end objective encourages the minimization of the KLD, some latent dimensions typically contribute near zero information. We identify these dimensions by Monte Carlo estimating the KLD induced by each latent bit using source samples; any dimension contributing less than $\frac{1}{1000}$th of the total KLD is thresholded and removed. This ensures that the channel simulation resources are prioritized for the most informative latent bits.

The operational performance of SBC is governed by the channel simulation block length $N$ (see Sec. 2.2 and 3.3). In our experiments, we set $N = 2^{23}$ ($\approx 8.4 \times 10^6$) to ensure sufficient polarization. For the arithmetic coding part in the `Compress` and `Decompress` routines, we utilize 50 Monte Carlo steps to estimate probabilities, and our final rate–distortion metrics are averaged over 10 independent simulation runs. As predicted by the theoretical result in Appendix C, we observed that the gap between training and operational performance decreases as $N$ increases, allowing the operational scheme to more closely match the performance during training. Despite the large block length, the $O(N \log N)$ complexity of the channel simulation scheme yields an operational scheme that remains computationally efficient.

To support reproducibility and further research, we will release the source code upon publication.

