# OpenReview forum: "SoftBinary Coding: A New Information-Theoretic Paradigm for Neural Compression via Fast Channel Simulation"
_ICML.cc/2026/Conference — ICML 2026 regular_

### Official Review · Reviewer_1jrx · 2026-03-04

**Soundness:** 1
**Presentation:** 1
**Significance:** 2
**Originality:** 2
**Overall Recommendation:** 2
**Confidence:** 4

**Summary:**

The authors considered combining neural coding, channel simulation, and polar codes for neural compression. The motivation was that non-linear transform coding approximates the coding rate using a surrogate, which may not correctly reflect the true coding rate. The hope was by taking the approach of channel simulation, this issue could be resolved. The authors proposed a system that can use polar coding, and showed that the performance is comparable or even better than TCQ.

**Compliance With Llm Reviewing Policy:**

Affirmed.

**Final Justification:**

After careful consideration, I decided not to raise the score (I was debating whether to raise it before the authors' second round of reply, but their reply convinced me not to). The authors need to improve the quality of the presentation significantly. There are too many issues in the presentation (both the logic flow and the details), which made the reading a frustrating experience. I'm not sufficiently convinced of the correctness due to these issues. I feel the underlying idea can be interesting, but the authors need to reorganize the logic flow and carefully proofread their paper; only then can an assessment of the work be made reliably.

**Key Questions For Authors:**

See the comments given in the weakness.

**Limitations:**

I didn't find a discussion on the limitations.

**Strengths And Weaknesses:**

Strengths:
1. The components used are certainly non-trivial, and the idea is interesting.
2. The experimental results seem to be good. TCQ is a technique that not many people are aware of, and it is nice to see that the authors use it as a baseline.

Weaknesses: One major issue is that the paper is poorly written, and I believe the paper is missing a significant part regarding the compression rate in its relation to polarization. There is no clear description of the coding procedure except in the algorithms. This likely have lead to many issues, some of which are fatal. Though I see some merits in the underlying idea (if I guess their idea correctly), there are simply too many confusions/issues in this version, and it is impossible to even determine the correctness or completeness of the work.

1. The coding rate is roughly the entropy of \Delta^N. However, there is no analysis of this rate. Instead, the authors studied the polarization behavior of the U_i given U^{i-1}, V^N, \Pi. There is not necessarily a connection between them. If the connection is not needed, then it seems the polarization behavior is not needed either, and we can use any u and \tilde{u}, and just transmit the difference.
2. It is not clear what the authors really mean by polarization: Above Theorem 3.1, the authors stated that P(U_i=1|U^{i-1}, V^N, \Pi) goes to 0 and 1, and this means polarization. This is not the true meaning of polarization in the sense of Arikan's polar code. The theorem does state that the entropy goes to either 0 or 1 almost surely, instead, which would be closer to the correct meaning.
3. Is the polarization of  H(U_i=1|U^{i-1}, V^N, \Pi) what we need? Since the decoder does not have V^N, is this even related to the compression rate? If not, what is the significance?
4. The legends in the figure are confusing: the dotted lines are missing their legends.
5. What is the meaning of one-shot coding in this context? If the authors are using block length =1, then it should be impossible for it to beat the one-shot R-D bound?
6. The figures are badly done. Figure 1 would suggest a single X is mapped to one V and one Z, which would limit the rate to less than 1 bit/per source symbol, but this should not be the case (to L bits instead?). Figure 2 is also confusing; the plot appears to suggest U^N is generated directly from V^N, but this should not be the case. Where is the decoder in Figure 2? These two figures caused significant issues during my reading, and it was rather frustrating.
7. Some key components are missing, particularly regarding the computation of P(U_i|U^{i-1}, PI).
8. One technical issue is regarding the Lyapunov function, which the authors claim decreases by a fixed amount at each step. Here, the complication is that the distribution is not i.i.d, and it would be very tricky to prove the same convergence result. I was not able to pinpoint the key new technique that allows the authors to overcome this difficulty. Note that when they are not identically distributed, this decrease would be dependent on the profile in a more complex manner.
9. What is the importance of the random permutation \Pi? If does not seem to be of key importance, but causes more issues. For example, in Figure 2, don't you also need it for the InversePolarEnc block?
10. Proof in the appendix seems to state a result in expectation, instead of the convergence in Theorem 1. A step seems to be missing.
11. Since the coding rate is roughly H(\Delta^N), and the training is essentially using the mutual information as its surrogate, isn't this still using a surrogate, simiarlyly as NTC?

---

> ### Author Rebuttal · Authors · 2026-03-31
>
> We thank the reviewer for their evaluation. For a discussion on the limitations, we refer them to Section 6 (lines 406-421). Our responses to the identified weaknesses are provided below:
> 1. The rate of the scheme is characterized in Appendix C, where we establish its first-order optimality. We refer to this in the text at lines 254-260. Specifically, (82) bounds the operational rate in terms of the subchannel conditional entropies, and (86)-(111) show this bound is asymptotically optimal as the subchannel entropies polarize. At a high level, polarization is what makes the correction sequence $\Delta^n$ compressible. Without polarization, the scheme would still simulate the target channel, but would incur a larger communication cost.
> The reviewer’s comment points to a way to improve the exposition: we plan to provide the statement of Thm. C.1 in the body of the paper.
> 2. The informal description of polarization preceding Thm. 3.1 is indeed missing a case. Refer to our answer to Q3 of Reviewer wDmK. Note that Thm. 3.1 is not, however, an almost sure convergence statement. Indeed, the sequence that is converging is not random, as the conditional entropies $H(U_i|U_1^{i-1}, V^N, \Pi)$ are expectations by definition.
> 3. We do indeed need $H(U_i=1|U^{i-1}, V^N, \Pi)$ both with and without the $V^N$ sequence: the rate is controlled by the difference between these two. But since the $V^N$ sequence is arbitrary, the result with the $V^N$ implies the result without it by taking $V^N$ to be deterministic. Please refer to the answer to Q1 for the role of polarization.
> 4. We explain the dotted lines in the text (refer to Sec. 5, lines 295-297), but omit this information from some of the legends due to crowding. We will revise the captions to make this clearer.
> 5. We use "one-shot" in the standard sense, namely that the encoder compresses a single instance of the source at a time. For a detailed description of the problem setting in the context of the circle and ramp, see Bhadane et al. (2022).
> The softbinary scheme is, strictly speaking, not one-shot; the channel simulator operates on many copies to realize shaping gains and thus obtains rates below the entropy-distortion bound. The neural network, on the other hand, operates on one source realization at a time. This is analogous to transform coding techniques such as NTC.
> 6. We apologize that the figure was frustrating to parse. This is partly because the scheme brings together ideas developed in different communities with different exposition styles and notation. We generally follow standard ML conventions, but we switch to information theoretic notation for our development of polarization (Footnote 1 marks this transition):
>
>     i) Consistent with ML conventions, in Fig. 1, $X$, $U$, and $V$ are allowed to be vectors.
>
>     ii) The polar decoder is implicitly present in the sampling blocks that generate $U_1,...,U_N$. It is a recursive algorithm that uses the polar structure internally. This is akin to how the polar transform is implicit in the successive cancelation decoder in polar (channel) codes.
> 7. In Sec. 3.3, lines 207-218 explain how the subchannel probabilities are computed using Arikan’s polar decoding algorithm. We felt that this algorithm is sufficiently well-known and intricate that a full exposition would add length without improving clarity. Our level of detail also matches the original PolarSim paper. Thus, we believe the current presentation strikes the right balance. That said, we would be happy to incorporate any specific suggestions the reviewer may have.
> 8. Non-stationarity of the inputs indeed complicates the analysis, as standard polarization proof techniques cannot be applied directly. We therefore adopt a different approach:
>
>     i) Rather than relying on martingale convergence, we construct a Lyapunov function that decreases by at least a fixed amount for any non-polarized channel. Crucially, this decrease does not depend on the degree of polarization (refer to Lemmas B.4 and B.5).
>
>     ii) Before each recursive polarization stage, we permute the indices. This symmetrizes the analysis and allows us to handle the non-stationarity of the input distributions.
> 9. (Cont'd from Q8). As noted in the paper, we omit the random permutations in practice as a satisfactory level of performance is achievable without them.
> The permutation seed $\Pi$ should indeed go into the InversePolarEnc block. We thank the reviewer for bringing this to our attention.
> 10. We assume the reviewer is referring to Thm. 3.1. As noted in our response to Q2, this is a statement regarding convergence of expectations rather than random variables.
> 11. The key difference is that with channel simulation, the operational rate of the scheme asymptotically approaches the training objective (see Thm. C.1) as the block length increases. This is not the case with standard NTC training methods where the mismatch between the two regimes persists even in the large block length limit.

---

> > ### Author Rebuttal · Reviewer_1jrx · 2026-04-03
> >
> > The rebuttal addressed some of my concerns, but due to the poor writing in the original submission, I still found it difficult to assess its correctness and significance. The overall logic of the method was not presented clearly. A few questions not clearly answered in the rebuttal are as follows.
> >
> > 1. The authors replied that "the rate of the scheme is characterized in Appendix C". Since this is perhaps the most significant result for the overall scheme, it should have been in the main text in the first place. However, it is still not clear what the authors are measuring here. Where was the notation \ell(b) defined, and where does its relation with H(\Delta_i) come from? Secondly, should we consider P(\Delta_i=1) or P(\Delta_i=1| \Delta_j, j<i)? Do you need the latter for the equality in (71)? This should be consistent with the measurements used in the empirical study.
> >
> > 2. I cannot quite understand what the reply regarding "$H(U_i=1|U^{i-1}, V^N, \Pi)$". Which Q1 were the authors referring to?
> >
> > 3. Regarding the proof for the non-stationary case. The authors pointed to Lemma B.4 and B.5. However, Lemma B.4 is problematic by itself: the constant C(\delta,\gamma) depends on the parameter \delta,\gamma, yet the parameter \gamma never showed up in the other part of the lemma statement, so it can be viewed as unconstrained. It was kind of "defined" above the lemma, but what x should be used in Lemma 4 precisely? It is then not clear whether the application of Lemma B.4 in (52-53) is correct. Is this quantity uniformly bounded by the same parameter?
> >
> > Overall, I believe the paper was put together in haste. Though it can be mostly correct, the sloppiness of the presentation does not allow a reasonable assessment of correctness, and it is not yet ready for a top-tier conference. A complete rewrite and a careful new round of review appear necessary, and the authors are encouraged to take significant effort in organizing the components and logical steps (both ML and information theory). It is perhaps not an easy task, but it is definitely necessary.

---

> > > ### Author Response · Authors · 2026-04-05
> > >
> > > We thank the reviewer for their timely reply. We are happy to hear that we could provide satisfactory answers for some of their questions. In response to their new questions:
> > >
> > > 1.  > The authors replied that "the rate of the scheme is characterized in Appendix C". Since this is perhaps the most significant result for the overall scheme, it should have been in the main text in the first place.
> > >
> > > It is debatable which of the two results is the most significant. Thm. C.1 establishes the overall optimality of the scheme, but most of the technical novelty lies in Thm. 3.1 in handling the non-stationarity of the source, as the reviewer themselves acknowledged this in their first set of comments:
> > >
> > > > Here, the complication is that the distribution is not i.i.d, and it would be very tricky to prove the same convergence result.
> > >
> > > Once Thm. 3.1 is established, the proof of Thm. C.1 is more standard. On the balance, and after some reflection, we agree that Thm. C.1 should be included in the body of the paper, as we had indicated in our earlier response. But we believe that the decision to highlight Thm. 3.1 in the original submission was *not* unreasonable or a sign of “*haste*” or “*sloppiness*”.
> > >
> > > > However, it is still not clear what the authors are measuring here. Where was the notation \ell(b) defined…
> > >
> > > The use of $\ell(\cdot)$ to denote the length of binary strings is a standard convention in information theory. We also explicitly define this in Sec. 2.2:
> > >
> > > > Our goal is to design a scheme, consisting of the common randomness, the encoder, and the decoder, which minimizes the average amortized length of the encoder's message $\frac{\ell(\mathsf{Enc}(\cdot))}{N}$
> > >
> > > The notation $b$ for the compressed bit string is introduced in Algorithm 1, which reads "compressed bit string $b$...".
> > >
> > > > ... and where does its relation with H(\Delta_i) come from?
> > >
> > > They are related through Eq. (80). Since $b$ is the compressed version of $\Delta^N$, the average length of $b$ is upper bounded by the entropy of $\Delta^N$ with an additional constant overhead. It is a fundamental result in lossless compression that any near-optimal prefix code, like arithmetic coding, satisfies this.
> > >
> > > > Secondly, should we consider P(\Delta_i=1) or P(\Delta_i=1| \Delta_j, j<i)? Do you need the latter for the equality in (71)?
> > >
> > > The equality in Eq. (71) is a simple application of the law of total probability. As such, it holds with both conditional and unconditional probabilities. We intended it as written.
> > >
> > > We upper bound the joint entropy $H(\Delta^N)$ by $\sum\limits_{i=1}^NH(\Delta_i)$. This sum is characterized by the marginal quantities $\Pr(\Delta_i)$ alone.
> > >
> > > 2. > I cannot quite understand what the reply regarding $H(U_i=1|U^{i-1}, V^N, \Pi)$. Which Q1 were the authors referring to?
> > >
> > > Instead of repeating the answer, we had to refer to a previous reply due to the character limit.
> > >
> > > We meant the first point in the list of weaknesses raised by the reviewer in their first comment posted, specifically:
> > >
> > > > ... it seems the polarization behavior is not needed either, and we can use any u and \tilde{u}, and just transmit the difference.
> > >
> > > To which our response was:
> > > > ...polarization is what makes the correction sequence $\Delta^n$ produced by Algorithm 1 compressible. Without polarization, the scheme would still simulate the target channel, but would incur a larger communication cost. This intuition is discussed at length in the original $\texttt{PolarSim}$ paper (Sec. 3.1, Sriramu et al.).
> > >
> > > 3. > Regarding the proof for the non-stationary case. The authors pointed to Lemma B.4 and B.5. However, Lemma B.4 is problematic by itself: the constant C(\delta,\gamma) depends on the parameter \delta,\gamma, yet the parameter \gamma never showed up in the other part of the lemma statement, so it can be viewed as unconstrained.
> > >
> > > The parameter $\gamma$ also appears in Lemma B.4 through the function $\phi$, which is defined in the line immediately before the lemma (which should read $\phi(x) = x^\gamma$).
> > >
> > > > It was kind of "defined" above the lemma, but what x should be used in Lemma 4 precisely? It is then not clear whether the application of Lemma B.4 in (52-53) is correct.
> > >
> > > The role of $x$ is played by the Bhattacharyya parameters in Eq. (41). In case the reviewer meant to ask about $\gamma$ instead of $x$, all our results hold for any value of $\gamma$ in $(0,1)$. The reader can fix any such value before Lemma B.4.
> > >
> > > > Is this quantity uniformly bounded by the same parameter?
> > >
> > > It is not clear which quantity the reviewer is referring to. In the proof of Lemma B.5, Eq. (53) does indeed follow from Eqs. (52) and (51) by applying the uniform bound in Lemma B.4 to each of the channel pairs defined in Eq. (49).

---

### Official Review · Reviewer_v7Fv · 2026-03-07

**Soundness:** 4
**Presentation:** 3
**Significance:** 4
**Originality:** 3
**Overall Recommendation:** 5
**Confidence:** 4

**Summary:**

The paper introduces SoftBinary Coding (SBC), a neural compression framework that replaces the standard quantization and entropy coding pipeline used in Nonlinear Transform Coding (NTC) with binary channel simulation.

The main technical contribution is a generalized PolarSim algorithm that supports independent, non-identically distributed binary channels with non-uniform marginals. The authors prove asymptotic rate optimality and show that the resulting scheme has O(N log N) computational complexity due to the polar coding structure.

The model is trained end-to-end using stochastic gradient estimators for discrete variables. Experiments on several synthetic information-theoretic benchmarks show that SBC can match or outperform NTC and classical quantization methods in some regimes.

**Compliance With Llm Reviewing Policy:**

Affirmed.

**Final Justification:**

The authors’ responses adequately address my concerns. The work remains technically strong, well-motivated, and novel, with clearly acknowledged limitations regarding scalability and real-world evaluation.

**Key Questions For Authors:**

1. Have the authors attempted to apply SBC to real datasets such as images (e.g., standard image compression benchmarks)?

2. The paper mentions increased difficulty when training on higher-dimensional sources. How does the method scale as the latent dimensionality grows, and what are the main limiting factors (optimization, memory, or channel simulation cost)?

**Limitations:**

yes

**Strengths And Weaknesses:**

# Strengths

The paper is well motivated from an information-theoretic perspective and clearly highlights limitations of quantization-based pipelines as well as challenges in existing channel simulation approaches. It proposes a new algorithm for binary channel simulation based on a generalized PolarSim construction that supports independent, non-identically distributed channels with non-uniform marginals. The method is theoretically supported and achieves O(N log N) complexity due to the polar coding structure, improving scalability compared to earlier channel simulation approaches with exponential complexity.

The experimental design is consistent with the theoretical focus of the work. The use of synthetic sources with known rate–distortion limits allows meaningful comparisons with optimal bounds and helps isolate the behavior of the proposed method.

The paper is clearly written and well organized, and it positions the work appropriately relative to existing literature on neural compression, quantization, and polar coding. The overall idea of replacing the standard transform–quantize–entropy coding pipeline with stochastic binary latent variables generated through binary channel simulation provides an interesting alternative perspective on neural compression.

# Weaknesses

The evaluation is limited to low-dimensional synthetic sources. It remains unclear whether the method can be applied effectively to realistic data such as images, audio, or video, and experiments on real datasets would significantly strengthen the paper.

The scalability with respect to latent dimensionality is also unclear. The paper briefly mentions increased training difficulty at higher dimensions, similar to the curse of dimensionality observed in vector quantization, but this limitation is not analyzed in detail. Since practical compression tasks require much higher-dimensional latent spaces, this aspect deserves further discussion.

Finally, the claim that NTC performs poorly at ultra-low bitrates due to quantization limitations is not strongly supported by the presented experiments. Additional empirical evidence would help substantiate this argument.

---

> ### Author Rebuttal · Authors · 2026-03-31
>
> We thank the reviewer for their careful assessment of our work. Our responses to the identified weaknesses and key questions are provided below:
>
> ### **Weaknessess**
> > Finally, the claim that NTC performs poorly at ultra-low bitrates due to quantization limitations is not strongly supported by the presented experiments. Additional empirical evidence would help substantiate this argument.
>
> We agree that there is not substantial empirical evidence supporting the theory that NTC performs poorly at low bit rates specifically due to its quantization limitations. Indeed, it is not one of the scientific claims of our paper. That said, the theory is supported by the following reasoning:
>
> Let us first clarify that entropy-constrained scalar quantizers such as NTC are generally subjected to a ~1.53 dB distortion penalty ("shaping gain") in the high-rate regime (Gish & Pierce, 1968 [1]). Empirically, this is evident in all rate–distortion plots in the paper – the rate–distortion curve of NTC is consistently above the SBC curve, and aligns with the solid black line (Figures 3 and 4) and the dashed black line (Figure 7), which correspond with this result. For the sources in other figures, such an analytical result is not available, but the existence of a similar bound can be imagined based on the asymptotic behavior of the NTC curve.
>
> In the low-rate regime, we cannot characterize the penalty on scalar quantizers precisely. However, from the RD plots in Figures 5, 6 and 7, we can see that NTC generally performs worse than SBC in the low-rate regime as well, and in fact worse than any other method we show that is not based on scalar quantization. In Figures 3 and 4, NTC performs analogously, but due to space constraints we decided to omit the low-rate regime. In Figure 8, the performance gap is largely due to the mismatch of the entropy model.
>
> In summary, NTC suffers from a significant performance penalty in both low- and high-rate regimes. However, it should be noted that a constant performance gap in terms of dB tends to be more prominent in terms of signal-to-noise ratio at low rates, as dB is a logarithmic measurement.
>
> [1] Gish, H., & Pierce, J. (1968). Asymptotically efficient quantizing. IEEE Trans. Inf. Theory. [link: https://ieeexplore.ieee.org/document/1054193 ]
>
>
> ### **Key Questions**
> 1-2. We have attempted training on images – our experience indicates that the application to natural images is limited mainly by the training time, requiring a much longer time to converge. We are not able to complete training on images using the compute resources available to us, but this could in principle be done by other groups (also see our response to "Weaknesses" raised by the Reviewer nGRg). As indicated in the Discussion, we believe this is a promising direction for future work.

---

> > ### Author Rebuttal · Reviewer_v7Fv · 2026-04-02
> >
> > The authors’ rebuttal adequately addresses the concerns. The theoretical contribution is sound, novel, and well-motivated. The main limitations concern empirical validation on high-dimensional or real-world data, which the authors acknowledge transparently. The original recommendation of accept remains justified.

---

> > > ### Author Response · Authors · 2026-04-05
> > >
> > > We thank the reviewer for their timely reply. We are happy to hear that we could answer their questions about our proposed model’s scalability and its comparison with NTC in the low-rate regime. We are also pleased to hear that the reviewer found the paper novel, well-motivated, and technically solid.

---

### Official Review · Reviewer_wDmK · 2026-03-09

**Soundness:** 3
**Presentation:** 2
**Significance:** 3
**Originality:** 3
**Overall Recommendation:** 4
**Confidence:** 3

**Summary:**

The authors have proposed the soft binary coding framework as an alternative to Neural compression, with treating lossy compression as a channel simulation problem.
In their setup, the encoder produce parameters V, which defines a series of Bernoulli variables Z as the latent representations, and whose distribution given input X is determined by V. The decoder, sharing randomness with the encoder, tries to sample the variable Z according to the Bernoulli distribution P(Z|V) with minimum cost(optimizing the rate).
This is achieved using a variation of PolarSim, which polarize the channels $U_i \mid U_1, \ddots, U_{i-1}, V^N, \Gamma$, and hence entropy encoding can be utilized for this polarized channel with optimum compression rate.

**Compliance With Llm Reviewing Policy:**

Affirmed.

**Final Justification:**

While the work presents an interesting concept, the overall organization must be significantly improved to enhance clarity and facilitate further development in the field; therefore, I am maintaining my current score.

**Key Questions For Authors:**

The work has not an overall explanation for the pipeline, specifically, I wonder
1. How is the $\bar{p_i}$ or the clean sub-channels being derived. It is understood that it is determined by $\bar{p_i}$ which is the marginal distribution of $U_i$, and $P(U_i \mid U_\cdots, V^N, \Gamma)$, and how is this conditional probability estimated or calculated and being transmitted, as 'slight overhead’. A thoroughly framework introduction would clarify the structure better.
2. Why is polarization necessary in this scenario, and what advantages does it provide over naïve entropy encoding for sampled Z? While the authors mention that “Optimally compressing data such as sparse point cloud attributes may benefit from more flexible schemes,” this seems to implicitly suggest a key advantage of the proposed method. However, the link is not made sufficiently clear. The authors are encouraged to provide a more explicit discussion of how the proposed scheme offers greater flexibility and the specific benefits this brings for such data.
3. In line 185 and 186, perhaps the author mean the entropy instead of the probability, i.e., $H(U_i\mid \cdots)$ instead of $P(U_i=1\mid\cdots)$, if consistency with Theorem 3.1 were to be preserved.
4. Why does the experiments section reports NTC with lower latent dimension L value than SBC in most of the cases?
5. What is the computational complexity of baseline methods, especially NTC which is the main baseline? The authors only present the complexity of the proposed approach. Inference-time comparisons (e.g., practical runtime) would also be valuable for a clearer assessment.

**Limitations:**

yes

**Strengths And Weaknesses:**

Strengths:
The mathematical induction is sound and valid.
The proposed method addressed the non-differentiability during quantization, smoothness bias, for the existing NTC framework, as well as having better Rate-Distortion tradeoff, as being examined in their experiments.

Weakness:
The structure and framework of this work is rather unclear, partly due to the page limit. It is adviced that the author provide a clear pipeline description for the derivation/sampling of the variables at each stages.
The application of the method only focus for Bernoulli iid sampling with toy examples, although achieving benefit over NTC, it has not demonstrate its potential application in more complicated scenarios.
The scheme proposed is unstable at training and relies on empirical intervention even at toy examples.
The system is complex and requires extremely large block size.

---

> ### Author Rebuttal · Authors · 2026-03-31
>
> We thank the reviewer for their thoughtful feedback. Our responses to the identified weaknesses and questions are provided below:
>
> ### **Weaknesses**
>  Our overall scheme reduces an arbitrary compression problem to one that involves simulating a binary-output channel. As such, it is not restricted to binary setups in any way, as evidenced by our experiments. Crucially, **binary-output channels in question do not, in fact, need to be i.i.d.**, which is a key technical contribution over $\texttt{PolarSim}$ (Sriramu et al., 2024); see Sec. 3, lines 136-154.
>
> We clarify that the scheme is not "unstable" but exhibits initialization dependence, a common trait in discrete latent models like VQ-VAE and $k$-means. No manual interventions were required for the sources evaluated; all results are reproducible as shown. For further discussion on initialization, block length, and natural images, please see our response to Reviewer nGRg.
>
> ### **Key Questions**
>
> 1. Unlike the channel coding use of polar codes, which requires an explicit demarcation of "frozen" subchannels, our scheme treats both low-noise and high-noise subchannels in a unified manner. The rate contribution of each subchannel is determined by the pmf $\Pr(\Delta_i = 1)$, which serves as the probability table for arithmetic coding.  In Sec. 3.3., we note that:
>
> > We use arithmetic coding (Rissanen, 1976)  with a probability table that is obtained via Monte Carlo estimation of $\Pr(\Delta_i = 1)$.
>
> Since this distribution is input-independent, it can be computed offline and made available to both the encoder and decoder, incurring no communication overhead. The overhead we refer to instead arises from estimation error: if the probability table used by the arithmetic coder does not match the true pmf of the data being compressed, a rate penalty is incurred. This penalty can be reduced by improving the accuracy of the Monte Carlo estimate.
>
> 2. Please refer to our answer to Q1-2 from Reviewer 1jrx for the role of polarization in our scheme. With regards to flexibility, we are referring to the fact that our scheme, similar to VQ, is able to achieve a "shaping gain" over more constrained entropy-coded scalar quantization methods such as NTC (a more detailed discussion of this is in our response to Reviewer v7Fv). In terms of signal-to-noise ratio, a shaping gain of a certain dB tends to be much more prominent in the low-rate regime, which is why we think our method may show benefit in coding of data that is low-dimensional and low-rate, such as attributes of point clouds (as compared to images, which are very high-dimensional).
>
> 3. There is a typo in our description which likely contributes to the confusion. We thank the reviewer for bringing this to our attention. The correct statement is:
>
> > .. we have $\Pr\left(U_i = 1 \mid U^{i-1}, V^N, \Pi\right) \approx 0$ or $\Pr\left(U_i = 1 \mid U^{i-1}, V^N, \Pi\right) \approx 1$ or $\Pr\left(U_i = 1 \mid U^{i-1}, V^N, \Pi\right) \approx \frac{1}{2}$.
>
> Since $U_i$ is a Bernoulli random variable, its entropy is maximized when it is equally likely to take both possible values. Likewise, if it is heavily biased towards either possibility, its entropy diminishes.
>
> 4. This is due to the fact that latent dimensions in NTC vs. SBC are subject to different constraints. In SBC, one latent dimension corresponds to one (_stochastic_) bit, and thus the upper limit on the information that can be transmitted using SBC is 1 bit per dimension. In NTC, as each dimension represents a scalar on the real line (or a signed integer, after quantization), there is no such upper limit. However, NTC inherently appears to have a bias for continuity in the latent space, which SBC does not, and may require more complex priors ($p$) for good performance, while SBC simply uses a Bernoulli $p$. It is therefore expected (and in fact part of the design) that SBC will need a larger number of latent dimensions for a similar rate–distortion performance.
>
> 5. In general, comparing runtime can be a difficult undertaking, as it depends on the design of the hardware used, and on the maturity of the software implementation. As our method uses a first-of-its-kind algorithm, we believe empirical comparisons aren't really useful at this stage, as we haven't taken steps to bring our algorithm to a similar level of maturity as arithmetic coding implementations. For now, we see the fact that our algorithm runs with $O(N \log N)$ complexity as a major advantage, as opposed to existing general channel simulation schemes, which have exponential complexity in block length.
>
> Another factor is that the overhead of arithmetic coding tends to be so low that we gave NTC the benefit of the doubt and simply reported cross-entropy estimates (which tend to be slightly better than the real coding rate). If we wish to compare runtime empirically, we would need to make a decision regarding which arithmetic coding implementation to use, which can have a measurable effect on the result.

---

> > ### Author Rebuttal · Reviewer_wDmK · 2026-04-02
> >
> > Thanks for the authors' responses and they have addressed all my questions.

---

> > > ### Author Response · Authors · 2026-04-05
> > >
> > > We thank the reviewer for their timely reply. We are happy to hear that we answered their questions regarding sub-channel probability estimation, polarization, choice of latent dimensionality for NTC, and computational complexity.
> > >
> > > Given this, we remain hopeful that the reviewer will consider revising their score in their final evaluation of the paper.

---

### Official Review · Reviewer_nGRg · 2026-03-13

**Soundness:** 3
**Presentation:** 3
**Significance:** 3
**Originality:** 3
**Overall Recommendation:** 4
**Confidence:** 1

**Summary:**

This paper proposes SoftBinary Coding (SBC), a neural compression framework that replaces the prior NTC with a stochastic binary latent representation and channel simulation. Instead of mapping inputs to continuous latents followed by non-differentiable quantization, the proposed method encodes data into binary latent variables and realizes the encoder distribution through a fast channel simulation scheme based on polar codes.

**Compliance With Llm Reviewing Policy:**

Affirmed.

**Key Questions For Authors:**

The proposed method repeats each optimization with three different random initializations and selects the one with the lowest loss. It would be helpful if the authors could clarify why the proposed training procedure appears to depend strongly on initialization.

**Limitations:**

yes

**Strengths And Weaknesses:**

### strengths

- The main strength of this paper lies in its theoretical contribution is clear. While PolarSim originally operates under strong assumptions that may be difficult to satisfy in practical settings, this work extends the framework to a more realistic regime and theoretically establishes both a generalized polarization theorem and the first-order optimality of generalized PolarSim.
- The paper demonstrates, through both theoretical analysis and experiments, that the proposed SBC framework can mitigate several limitations of NTC, such as train–test mismatch, smoothness bias, and the lack of shaping gain.

### weaknesses

- Its practical effectiveness from the perspective of neural compression is somewhat unclear. The paper uses a very large block length of $N=2^{23}$ to obtain sufficient polarization. This may imply the need for a very large buffer, which could ultimately translate into a buffering-delay issue in real-time transmission scenarios.
- The experimental evaluation is conducted only on idealized information-theoretic sources. While these benchmarks are useful for analyzing theoretical properties, it remains unclear how the proposed method performs on more realistic data distributions.
- Although this is mentioned as future work, the paper provides limited comparisons on sources where NTC is already known to perform well (e.g., natural images with relatively smooth distributions).

---

> ### Author Rebuttal · Authors · 2026-03-31
>
> We thank the reviewer for their thoughtful comments and the time spent evaluating our work. Our point-by-point responses to the identified weaknesses and key questions are provided below:
>
> ### **Weaknesses**
>
> 1. The block length is _not_ as large as it might appear. The block length quoted in Appendix D refers to the number of binary-output channels to be simulated. The block length in terms of the number of latent variables, which is arguably the more meaningful quantity, is lower by a factor of up to $L$ (depending on the outcome of the latent pruning process described in Section 4.2). Also, in the context of channel coding, the original polar codes did not have competitive short-to-moderate block length performance, but this was rectified with later optimizations. The proposed scheme is akin to the original polar code but now applied to channel simulation. We likewise expect that the short-to-moderate block length performance of this scheme can be improved, although these improvements are outside the scope of the paper. Finally, it can be viewed as a virtue that the scheme can scale to such large block lengths and thereby reap the associated shaping gains; many practical channel simulation schemes in the literature have exponential complexity with respect to the block length.
>
> 2-3. Our experience indicates that the application to natural images is limited mainly by the training time. We are not able to complete training on images using the compute resources available to us, but this could in principle be done by other groups. Also, our scheme achieves state-of-the-art performance on the VQ problem for Gaussian sources (Figure 7). Thus, the proposed scheme could be practically useful as a plug-in replacement for a VQ module within other schemes (including NTC itself). In this sense, we would like to highlight that the sources considered are _not_ purely of theoretical interest.
>
>
> ### **Key Questions**
>
> We found empirically that the training procedure yields slightly different results after reinitialization. We also found that it is beneficial to use the regularization term mentioned in the paper (see the text below Eq. (9), i.e., lines 268-274 in Section 4.1), keeping the encoder near the decision boundary at the beginning of training, and gradually phasing it out.
>
> We have not determined the reason for this initialization dependence with certainty. However, it is not entirely unexpected. While we generalize entropy-coded quantization by simulating a stochastic channel, the channel is still discrete-output (_in fact_, binary). Learning autoencoder-like models with discrete latent variables often suffers from initialization dependence (for example, see VQ-VAE [1] and Gumbel-Softmax based models [2,3]). In fact, even classical algorithms such as vector quantizers trained with Lloyd's algorithm [4] / $k$-means clustering are well-known to suffer from initialization dependence. NTC appears to escape this, likely because its training objective represents a continuous proxy for the inherently discrete learning problem. Thus, one hypothetical reason for why this model is initialization dependent could be that its latents are still discrete. One direction of future work we are interested in is to develop a similar coding scheme but for continuous latents (potentially also based on polarization). It would be interesting to see whether the initialization dependence would carry over to such a model.
>
> [1] Van Den Oord, A., Vinyals, O., & Kavukcuoglu, K. (2017). Neural discrete representation learning. Advances in Neural Information Processing Systems (NeurIPS). [link: https://arxiv.org/abs/1711.00937 ]
>
> [2] Jang, E., Gu, S., & Poole, B. (2017). Categorical reparameterization with Gumbel-Softmax. International Conference on Learning Representations (ICLR). [link: https://arxiv.org/abs/1611.01144 ]
>
> [3] Maddison, C. J., Mnih, A., & Teh, Y. W. (2017). The concrete distribution: A continuous relaxation of discrete random variables. International Conference on Learning Representations (ICLR). [link: https://arxiv.org/abs/1611.00712 ]
>
> [4] Lloyd, S. (1982). Least squares quantization in PCM. IEEE Transactions on Information Theory. [link: https://ieeexplore.ieee.org/document/1056489 ]

---

> > ### Author Rebuttal · Reviewer_nGRg · 2026-04-03
> >
> > Thank you for the response. However, my concerns regarding Weaknesses #2 and #3 remain unresolved. Therefore, I will maintain my original score.

---

> > > ### Author Response · Authors · 2026-04-05
> > >
> > > We thank the reviewer for their timely reply and are glad to hear that our response answered their questions regarding block length and initialization dependency. As noted in the Discussion, we recognize that evaluating the scheme on high-dimensional data remains a limitation of the current work. We hope to validate our scheme through more extensive experiments on high-dimensional sources (e.g., images) in future work.
> > >
> > > We believe that the reviewer understands both the strengths and weaknesses of the paper well and that they would be merited to increase their confidence score.

---

### Decision · Program_Chairs · 2026-04-30

**Decision:**

Accept (regular)

**Comment:**

The reviewers agree that the idea is interesting and appreciated the theoretical analysis and the experimental comparisons with TCQ. At least two reviewers still have concerns about the paper, particularly the organization, the lack of an overall explanation for the pipeline, and key results such as Theorem C.1 being relegated the appendix. The concerns about clarity and organization are important, but on balance, my view is these issues can be addressed in the final version. I trust the authors to do so, and would also encourage them to: i) recall notation where necessary, e.g. when it is buried within an algorithms, and ii) make the limitations of the paper more explicit, e.g. include a discussion on why this scheme cannot yet be applied to high-dimensional sources like natural images. Carefully addressing all the issues raised by the reviewers will make the paper stronger and more easily readable.